# SAM homeostasis is regulated by CFIm-mediated splicing of MAT2A

Anna M Scarborough[1], Juliana N Flaherty[1], Olga V Hunter[1], Kuanqing Liu[2], Ashwani Kumar[3], Chao Xing[3,4,5], Benjamin P Tu[2], Nicholas K Conrad[1]*

[1]Department of Microbiology, UT Southwestern Medical Center, Dallas, United States; [2]Department of Biochemistry, UT Southwestern Medical Center, Dallas, United States; [3]Eugene McDermott Center for Human Growth and Development, UT Southwestern Medical Center, Dallas, United States; [4]Department of Bioinformatics, UT Southwestern Medical Center, Dallas, United States; [5]Department of Population and Data Sciences, UT Southwestern Medical Center, Dallas, United States

**Abstract** S-adenosylmethionine (SAM) is the methyl donor for nearly all cellular methylation events. Cells regulate intracellular SAM levels through intron detention of MAT2A, the only SAM synthetase expressed in most cells. The $N^6$-adenosine methyltransferase METTL16 promotes splicing of the MAT2A detained intron by an unknown mechanism. Using an unbiased CRISPR knock-out screen, we identified $CFI_m25$ (NUDT21) as a regulator of MAT2A intron detention and intracellular SAM levels. $CFI_m25$ is a component of the cleavage factor Im ($CFI_m$) complex that regulates poly(A) site selection, but we show it promotes MAT2A splicing independent of poly(A) site selection. $CFI_m25$-mediated MAT2A splicing induction requires the RS domains of its binding partners, $CFI_m68$ and $CFI_m59$ as well as binding sites in the detained intron and 3′ UTR. These studies uncover mechanisms that regulate MAT2A intron detention and reveal a previously undescribed role for $CFI_m$ in splicing and SAM metabolism.

*For correspondence: nicholas.conrad@utsouthwestern.edu

Competing interests: The authors declare that no competing interests exist.

## Introduction

S-adenosylmethionine (SAM) is the universal methyl donor, essential for the methylation of DNA, RNA, and proteins that regulates most cellular functions. Humans express a single SAM synthetase, encoded by the gene *MAT2A*, in nearly every tissue except the liver (*Murray et al., 2019*). The protein encoded by *MAT2A*, MATα2, uses ATP and L-methionine to produce SAM. Upon donating a methyl group, SAM converts to S-adenosylhomocysteine (SAH), which inhibits some methyltransferases (*Ferreira de Freitas et al., 2019*). Therefore, SAM must be constantly produced to maintain SAM-to-SAH ratios sufficient to support methyltransferase activity (*Clarke, 2006*; *Krijt et al., 2009*; *Walsh et al., 2018*). Too much or too little methylation of substrates by SAM has been associated with cancer, diabetes, along with other diseases (*Gharipour et al., 2020*; *McMahon et al., 2017*; *Zhu et al., 2020*). Therefore, intracellular SAM levels are tightly controlled to maintain homeostasis, with loss of regulation resulting in cellular dysfunction (*Gao et al., 2019*; *Lio and Huang, 2020*; *Ouyang et al., 2020*; *Parkhitko et al., 2019*).

We identified the $N^6$-methyladenosyl (m⁶A) transferase METTL16 as a key regulator of SAM-responsive MAT2A RNA splicing and proposed that it serves as a critical intracellular SAM sensor (*Pendleton et al., 2017*). METTL16 methylates MAT2A on six evolutionarily conserved hairpins (hp1-hp6) in the 3′ untranslated region (3′UTR; *Figure 1A*; *Parker et al., 2011*; *Pendleton et al., 2017*; *Warda et al., 2017*). These hairpins contain UAC*A*GARAA motifs which, in combination with their structure, allows for methylation by METTL16 at the A4 position (underlined)(*Doxtader et al., 2018*; *Mendel et al., 2018*; *Pendleton et al., 2017*). METTL16 interactions with MAT2A hairpins regulate

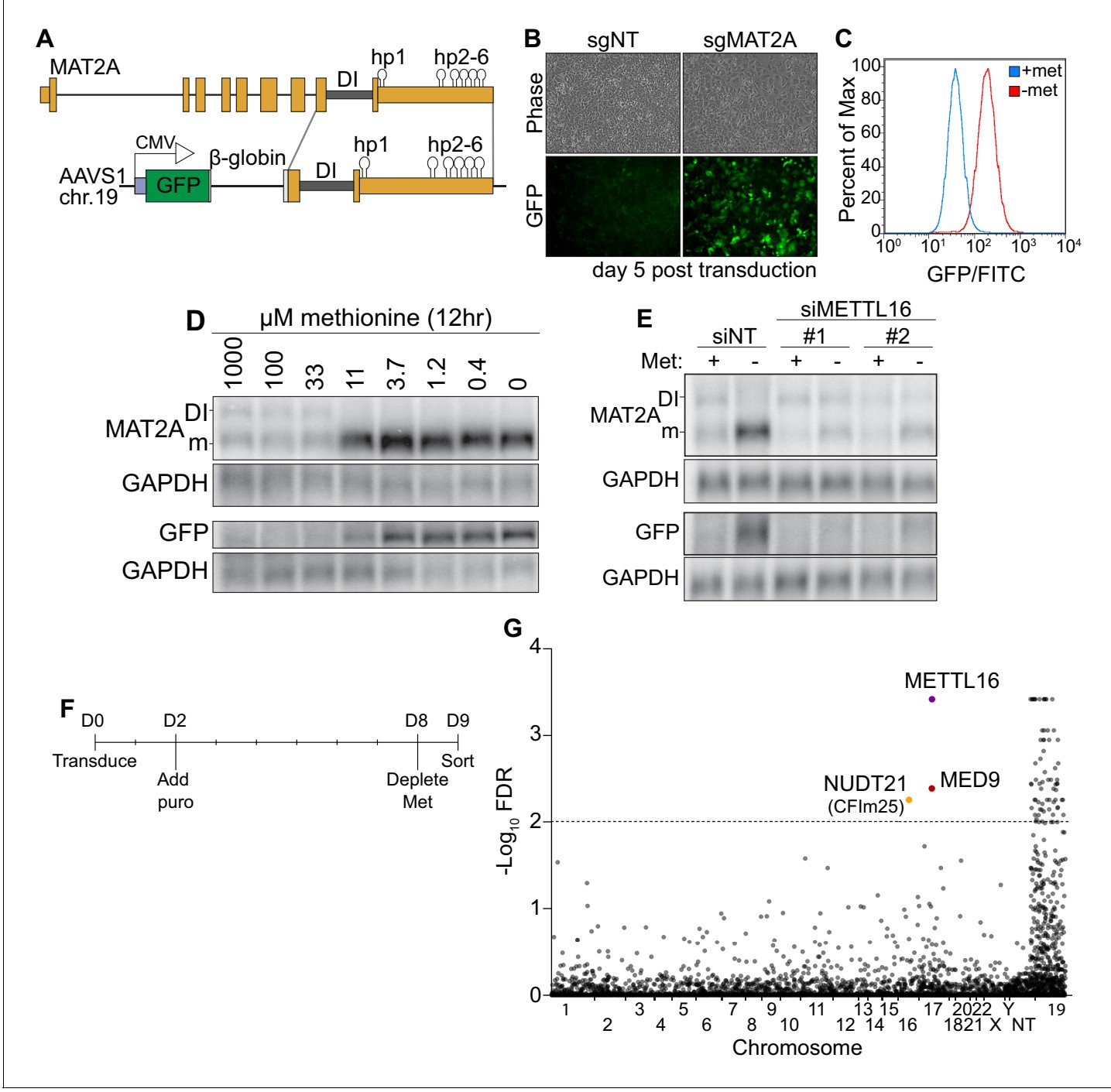

**Figure 1.** A CRISPR screen identifies CFI_m25 as a candidate MAT2A splicing factor. (**A**) Diagram of endogenous MAT2A gene (top) and GFP-MAT2A reporter (bottom). DI, detained intron; hp, hairpins; diagram is not to scale. (**B**) Representative images of reporter cells (phase) and GFP 5 days after transduction with lentivirus expressing Cas9 and sgRNA targeting MAT2A or non-targeting (sgNT) control. Cells were maintained in methionine-rich media. (**C**) Flow cytometry results monitoring GFP production in the reporter cell line after 24 hr conditioning in methionine-rich (blue) or methionine-free (red) media. Displayed as percent of maximum cell count for a given GFP intensity. (**D**) Northern blot analysis of GFP and endogenous MAT2A RNAs produced from the reporter line after 12 hr in media with the indicated methionine concentrations. In all figures, 'DI' and 'm' mark the MAT2A detained intron and mRNA isoforms, respectively. (**E**) Northern blot analysis of GFP and endogenous MAT2A RNA expression after four hours ± methionine depletion in the reporter line. Cells were treated with non-targeting control (siNT) or METTL16 (siMETTL16) siRNAs. (**F**) CRISPR screen timeline. (**G**) CRISPR screen results. CRISPR screen was performed in biological triplicate before analysis by MAGeCK. The -log_{10}(FDR) of the analysis is plotted on the y-axis and genes are organized alphabetically by chromosome number on the x-axis. Non-targeting (NT) guides are also included. Genes above the dotted line have an FDR < 0.01.

*Figure 1 continued on next page*

*Figure 1 continued*

The online version of this article includes the following figure supplement(s) for figure 1:

**Figure supplement 1.** An alternatively spliced reporter provides a more robust GFP signal.

MAT2A by two mechanisms, mRNA stability and pre-mRNA splicing. Upon methylation of hp2-6, the stability of the MAT2A mRNA decreases (*Martínez-Chantar et al., 2003*; *Pendleton et al., 2017*; *Shima et al., 2017*). In SAM-limiting conditions, reduced methylation of hp2-6 stabilizes the MAT2A mRNA. This mechanism likely reflects the typical 'reader-writer' paradigm in which m6A marks are read by m6A-specific binding proteins (*Meyer and Jaffrey, 2017*; *Yue et al., 2015*).

Here, we examine the mechanisms of a second function for METTL16 in regulating alternative splicing of the MAT2A transcript through its interactions with hp1. Our working model proposes that in SAM-rich conditions, METTL16 methylates hp1 then dissociates, resulting in increased retention of the last intron (*Figure 1A*; *Pendleton et al., 2017*; *Pendleton et al., 2018*). Failure to excise the last intron of MAT2A results in nuclear retention and degradation of the transcript, defining this intron as a detained intron (DI)(*Boutz et al., 2015*; *Bresson et al., 2015*; *Pendleton et al., 2018*). During SAM starvation, METTL16 binds hp1 but low SAM levels lead to decreased enzymatic turnover of METTL16. The resulting increased METTL16 occupancy on hp1 promotes splicing of the final MAT2A intron. Two carboxyl-terminal vertebrate conserved regions (VCR) in METTL16 are necessary and sufficient to promote splicing of MAT2A reporters and increase the affinity of METTL16 for RNA (*Aoyama et al., 2020*; *Pendleton et al., 2017*). Thus, with respect to hp1 and regulation of intron detention, METTL16 functions as both the m6A reader and writer. However, METTL16 lacks recognizable splicing domains and has no known protein interacting partners, so its mechanism for promoting splicing of the MAT2A DI has been unclear (*Ignatova et al., 2019*).

Splicing of terminal introns is coupled with 3′-end formation. Definition of both the 5′ and 3′ ends of an exon is necessary for efficient splicing, but terminal exons lack 3′ definition as there are no downstream introns (*De Conti et al., 2013*; *Martinson, 2011*; *Niwa et al., 1992*; *Niwa et al., 1990*). Instead, splicing factors and 3′ end formation machinery directly interact during the definition of the terminal exon (*Davidson and West, 2013*; *Dettwiler et al., 2004*; *Kyburz et al., 2006*; *Lutz et al., 1996*; *McCracken et al., 2002*; *Millevoi et al., 2006*; *Rappsilber et al., 2002*; *Shi et al., 2009*; *Vagner et al., 2000*). While splicing of terminal exons and 3′ end formation enhance each other, the mechanisms of this coupling remain incompletely understood.

The cleavage factor $I_m$ complex ($CFI_m$) is a component of the 3′-end formation machinery with characteristics that suggest it links 3′ end formation with splicing (*Hardy and Norbury, 2016*; *Martinson, 2011*). The $CFI_m$ complex consists of a dimer of $CFI_m25$ (NUDT21, CPSF5) and each $CFI_m25$ interacts with a monomer of $CFI_m68$ (CPSF6) or $CFI_m59$ (CPSF7) to form a heterotetrameric $CFI_m$ complex (*Kim et al., 2010*; *Rüegsegger et al., 1996*; *Rüegsegger et al., 1998*). The $CFI_m$ complex was identified as a 3′-end formation factor, but more recent work suggests it is not required for polyadenylation of all mRNAs. Instead, $CFI_m$ is an enhancer that regulates the efficiency of poly(A) site usage (*Zhu et al., 2018*). As a result, knockdown of $CFI_m$ components leads to widespread changes in polyadenylation, with the predominant effect being a shift to proximal poly(A) site usage (*Brumbaugh et al., 2018*; *Gruber et al., 2012*; *Li et al., 2015*; *Martin et al., 2012*; *Masamha et al., 2014*; *Zhu et al., 2018*). $CFI_m68$ and $CFI_m59$ contain arginine- and serine-rich regions (RS domains) that aid in 3′ end formation (*Zhu et al., 2018*). However, RS domains are common components of splicing factors, suggesting a role for $CFI_m68$ and $CFI_m59$ in splicing (*Dettwiler et al., 2004*; *Hardy and Norbury, 2016*; *Long and Caceres, 2009*; *Millevoi et al., 2006*; *Rappsilber et al., 2002*; *Rüegsegger et al., 1998*). Thus, in addition to its well-defined roles in 3′-end formation and alternative polyadenylation (APA), the $CFI_m$ complex may function in splicing. To date, there has been little direct evidence that any specific pre-mRNA requires $CFI_m$ as a splicing factor independent of its functions in poly(A) site selection.

Here, we uncover the role of the $CFI_m$ complex in METTL16-mediated splicing of MAT2A. Using a SAM-sensitive GFP reporter in a CRISPR knock-out screen, we identified $CFI_m25$ as a candidate factor in the regulation of MAT2A intron detention. Similar to loss of METTL16, knockdown of $CFI_m25$ results in a decrease in MAT2A mRNA after methionine, and therefore SAM, depletion. Additionally, knockdown of $CFI_m25$ results in a reduction of intracellular SAM levels in a detained intron

dependent manner. Although the CFI_m complex has a widespread role in alternative polyadenylation, it appears the regulation of SAM is separable from this function. We next identified two CFI_m25-binding sites that are necessary for the splicing of MAT2A. Finally, we show that CFI_m induction of splicing requires the RS domains of CFI_m68 or CFI_m59. This leads to an updated model of splicing for the SAM-synthetase MAT2A, in which METTL16 serves as an upstream SAM sensor that mediates splicing of MAT2A through the CFI_m complex.

## Results

### A CRISPR screen identifies CFI_m25 as a candidate MAT2A splicing regulator

To investigate the mechanism of METTL16-mediated splicing of MAT2A, we designed a SAM-responsive GFP reporter cell line. Our reporter construct consists of GFP, a β-globin intron with flanking exonic regions, the MAT2A DI with flanking exons, and full-length MAT2A 3′UTR driven by a CMV promoter (*Figure 1A*). We integrated the reporter into the AAVS1 safe harbor locus on chromosome 19 of HCT116 cells and isolated single cell clones (*Golden et al., 2017*; *Manjunath et al., 2019*; *Oceguera-Yanez et al., 2016*). In SAM-replete conditions, the MAT2A intron should be inefficiently spliced and produce little GFP protein, while SAM depletion should lead to efficient splicing, mRNA stabilization, and robust GFP signal. To validate our reporter line, we depleted intracellular SAM using CRISPR/Cas9 with an sgRNA targeting the endogenous MAT2A. As expected, MAT2A knockout increased GFP signal (*Figure 1B*). Moreover, depletion of methionine, and therefore SAM, robustly increased GFP as monitored by flow cytometry (*Figure 1C*) and western blot (*Figure 1—figure supplement 1A*, 'Reporter'). GFP-MAT2A mRNA accumulates upon shift to ~11 µM methionine media mirroring the endogenous MAT2A mRNA (*Figure 1D*). Similarly, knockdown of METTL16 abrogates the cell's ability to induce production of the endogenous MAT2A or reporter mRNA after methionine depletion (*Figure 1E*). Together, these observations demonstrate that the production of our reporter mRNA reflects that of the endogenous MAT2A.

Despite its usefulness in our studies (see below), we found that the reporter mRNA from our clonal line unexpectedly splices the β-globin 5′ splice site to the MAT2A 3′ splice acceptor (*Figure 1—figure supplement 1B–C*). Other isolated clonal lines that spliced as predicted produced less GFP protein upon methionine depletion, the majority of which was accumulated as a putative degradation product (*Figure 1—figure supplement 1A and D*, 'Predicted'). Since the reporter splicing creates a distinct C-terminal extension on GFP, we reasoned that the GFP C-terminal extension produced from β-globin and MAT2A exons 8 and 9 destabilizes the GFP reporter (*Figure 1—figure supplement 1B–C*). Indeed, treating cells with the proteasome inhibitor MG132 increased protein levels produced in lines that had the predicted splicing pattern, but GFP protein was unaffected in our reporter line (*Figure 1—figure supplement 1D*). Thus, the alternative splicing of the reporter line results in a more stable protein product and therefore more useful reporter. Nevertheless, our reporter robustly responds to intracellular SAM levels in a METTL16-dependent fashion, mimicking the endogenous MAT2A. Therefore, we used this reporter cell line in a screen for factors required to induce MAT2A expression upon methionine depletion.

To identify factors necessary for MAT2A induction, we performed a CRISPR knockout screen. We reasoned that if we knock out genes essential for induction of MAT2A splicing (e.g. METTL16), cells will have reduced ability to induce GFP expression upon methionine depletion. We transduced our reporter line with the puromycin-resistant lentiviral Brunello library, which contains four sgRNAs per protein-coding gene of the entire genome (*Doench et al., 2016*). Two days post-transduction, we added puromycin, selected over 6 days, then replaced media with methionine-free media. Eighteen hours later, we sorted for the lowest ~1% of GFP intensity cells (*Figure 1F*).

We identified seventy-two genes passing a 1% FDR cutoff from three independent biological replicates analyzed by MAGeCK (*Figure 1G* and *Supplementary file 1*; *Li et al., 2014*). However, nearly all 72 genes were found on chromosome 19, where the AAVS1 safe harbor locus resides. The simplest explanation for this overrepresentation is that we selected for recombination events that reduced GFP signal by removing the reporter. After exclusion of genes on chromosome 19, three hits remained. The top candidate was METTL16 which supports the efficacy of the screen. The other hits were MED9 and NUDT21 (CPSF5). MED9 is a member of the mediator complex, a coactivator of

RNA pol II (*Soutourina, 2018*). NUDT21 encodes the $CFI_m25$ protein. Given its association with RS domain-containing proteins $CFI_m68$ and $CFI_m59$, we decided to investigate potential functions for $CFI_m25$ in the regulated splicing of MAT2A.

## $CFI_m25$ is required for induction of MAT2A mRNA and maintaining SAM levels

To validate our CRISPR screen, we first examined the effects of knockdown of $CFI_m25$ on reporter RNA accumulation. Similar to METTL16 knockdown, we observed that $CFI_m25$ depletion resulted in lower mRNA accumulation (*Figure 2A–C and E*, 'Reporter') and a modest increase in the DI isoform (*Figure 2B and D–E*, and *Figure 2—figure supplement 1A*, 'Reporter'). Next, we tested the effects of $CFI_m25$ knockdown on a modified reporter. Into this reporter construct, we inserted a T2A 'self-cleaving' peptide after GFP to overcome the GFP destabilization by the β-globin/MAT2A C-terminal extension described above (*Figure 1—figure supplement 1*; *Luke and Ryan, 2018*). More importantly, we mutated hp2-6 thereby eliminating METTL16-dependent contributions to cytoplasmic stability of the mRNA isoform (*Figure 2A*, bottom). Using two independent clonal cell lines ('T2A, hp2-6m9#1' and 'T2A, hp2-6m9#2'), we observed decreases in mRNA accumulation and increases in intron detention in this reporter after METTL16 or $CFI_m25$ depletion (*Figure 2B–E* and *Figure 2—figure supplement 1*). These hp2-6-independent effects of $CFI_m25$ knockdown on reporter mRNA and DI isoform levels are consistent with a role for $CFI_m25$ in the induction of MAT2A splicing.

To confirm that $CFI_m25$ affects endogenous MAT2A and is not cell line specific, we knocked down $CFI_m25$ in 293A-TOA cells and analyzed MAT2A expression by northern blot analysis (*Figure 2F*). $CFI_m25$ depletion using two independent siRNAs increased intron detention under methionine-free conditions, phenocopying METTL16 depletion (*Figure 2F*, red). Interestingly, there was also a slight but significant increase in intron detention in methionine-replete media (*Figure 2F*, blue), further supporting a role for $CFI_m25$ in the regulation of MAT2A intron detention. $CFI_m25$ knockdown did not affect METTL16 expression (*Figure 2G*), so $CFI_m25$ does not regulate MAT2A expression by manipulation of METTL16 abundance. If METTL16 and $CFI_m25$ are both necessary for efficient splicing of the MAT2A detained intron, co-depletion should not exacerbate the effects of individual depletion of each factor. Indeed, co-depletion had no greater effects on MAT2A RNA than depletion of each individual factor (*Figure 2H*). Finally, we assessed SAM levels in cells depleted of $CFI_m25$ and found that, like METTL16, depletion of $CFI_m25$ decreased intracellular SAM levels under methionine-rich conditions (*Figure 2I*). Together, these data support the hypothesis that that $CFI_m25$ is necessary for production of SAM by regulating MAT2A splicing.

Knockdown of $CFI_m25$ results in global changes in poly(A) site usage. Indeed, a minor isoform of MAT2A using a weak, proximal poly(A) site in the 3′UTR has previously been detected (*Routh et al., 2017*). Therefore, we tested whether $CFI_m25$ regulates MAT2A by APA in our cells. The shorter isoform was not evident in our total RNA northern blots, so we poly(A) selected RNA for higher sensitivity. Two low-abundance bands of higher mobility were observed in 293A-TOA cells, either or both of which could be an APA product (*Figure 2—figure supplement 2A*). RNase H mapping results were consistent with one isoform being the previously reported APA isoform, but the longer isoform was of unclear origin (*Figure 2—figure supplement 2B*). Importantly, neither was responsive to methionine levels nor depletion of METTL16 or $CFI_m25$ (*Figure 2—figure supplement 2A*). APA patterns can be cell-type specific (see below), but these data support the conclusion that the observed changes in MAT2A expression and SAM levels in 293A-TOA cells (*Figure 2F and I*) are not due to $CFI_m$-mediated changes in MAT2A poly(A) site usage.

## $CFI_m25$ regulation of MAT2A requires the detained intron

Depletion of $CFI_m25$ alters poly(A) site selection on many transcripts, so the observed changes in MAT2A and SAM levels may result from processes unrelated to MAT2A intron detention. Conversely, it is formally possible that some of the changes in APA are secondary to the drops in SAM levels observed upon $CFI_m25$ depletion (*Figure 2I*). To test the importance of the DI, we used CRISPR to produce a clonal HCT116-derived cell line with the MAT2A DI deleted. Although we used a donor plasmid to promote homologous repair (HR) upon cleavage of two sites flanking the DI (*Figure 3A*), the resulting clonal line (116-ΔDI) is heterozygous. One allele contains the expected precisely deleted DI from HR while the second allele deletes the DI via non-homologous end joining

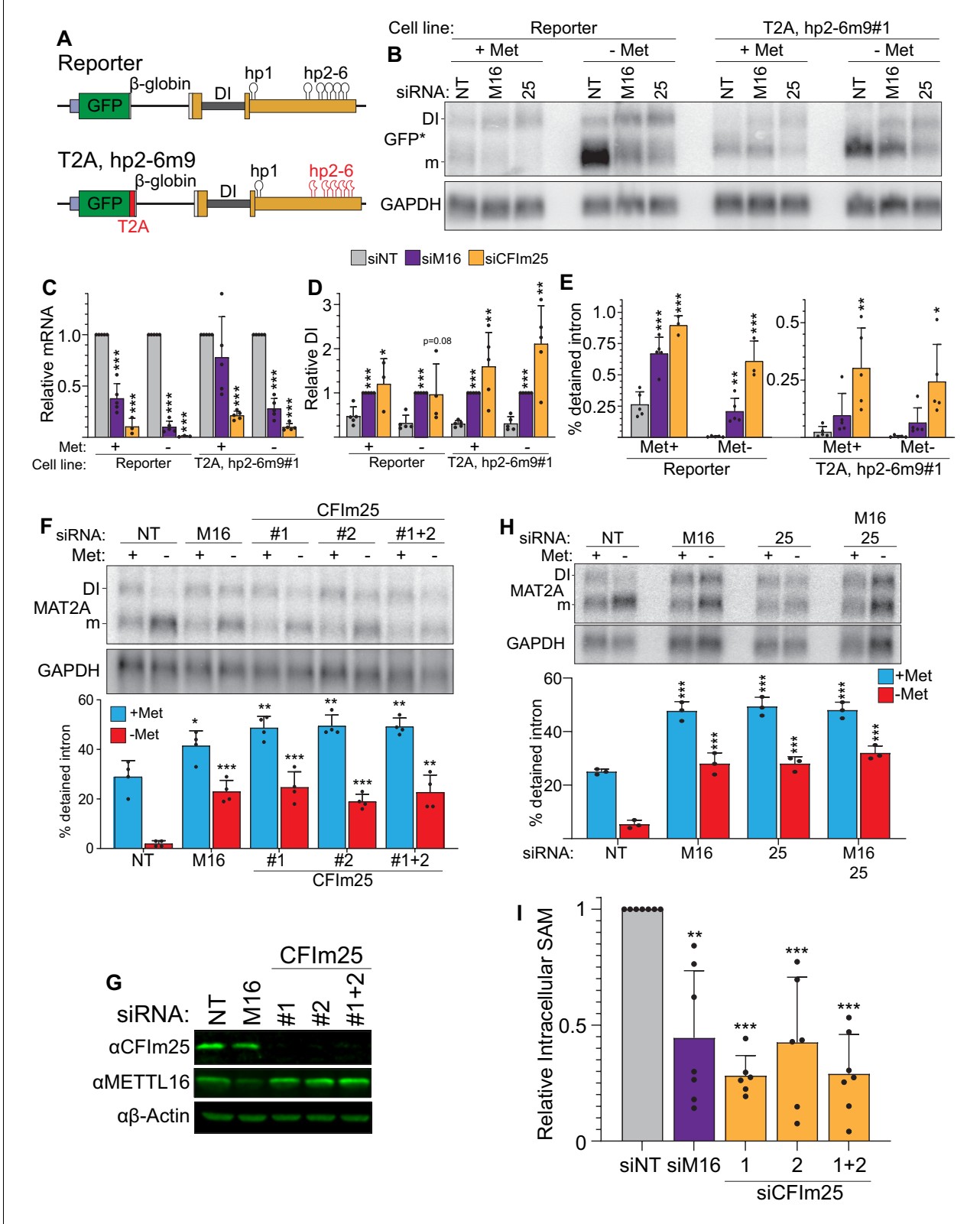

**Figure 2.** CFI_m25 regulates MAT2A splicing and activity. (**A**) Diagram of the original and hp2-6m9 mutant reporters. The latter reporter contains a T2A element (red) and nine point mutations (m9) in each of the hp2-6 (*Pendleton et al., 2017*). The modified reporter construct was inserted into the AAVS1 site of HCT116 cells and sorted to produce clonal cell lines. Diagram not to scale. (**B**) Northern analysis of GFP expression after knockdown with non-targeting (NT), METTL16 (M16), or CFI_m25 siRNAs in the original and modified reporter cell lines. Cells were conditioned with methionine-rich or -

*Figure 2 continued on next page*

*Figure 2 continued*

free media for 4 hr before harvesting. To increase signal-to-noise compared to preceding blots (*Figure 1*), we used poly(A)-selected RNA and developed an improved GFP northern probe (GFP*). Overexposed versions of this blot are included to more easily visualize the signal from the DI isoforms (*Figure 2—figure supplement 1A*). (**C and D**) Quantification of northern analyses as shown in panel B. Values were first normalized to GAPDH and are shown relative to siNT for the mRNA (**C**) or to siM16 for the DI isoform (**D**). Statistical analysis was performed relative to the siNT for both mRNA and DI isoforms. $n \geq 3$. (**E**) Quantification of the northern analysis in as in panel B expressed as percent detained intron. Statistical analysis was performed relative to the siNT. $n \geq 3$. (**F**) Northern analysis of MAT2A expression in 293A-TOA cells upon CFI$_m$25 or METTL16 knockdown. Two independent CFI$_m$25 siRNAs were tested. Cells were conditioned in methionine-rich or methionine-free media for 4 hr. $n = 4$. (**G**) Western analysis of CFI$_m$25 and METTL16 after the indicated knockdown in 293A-TOA cells. Actin serves as a loading control. $n \geq 3$. (**H**) Northern analysis of MAT2A expression after individual or co-depletion of METTL16 and CFI$_m$25. Knockdown proceeded for 4 days before conditioning cells in methionine-rich or methionine-free media for 4 hr. Quantified by percent detained intron. $n = 3$. (**I**) Intracellular SAM levels relative to non-targeting control after METTL16 or CFI$_m$25 knockdown in 293A-TOA cells. Statistical analysis compared all knockdowns to non-targeting control. $n \geq 6$. Unless otherwise noted, data are represented as mean ± SD and analyzed by a two-tailed, unpaired student's t-test compared to matched control. Significance is annotated as not significant (ns), *$p \leq 0.05$, **$p \leq 0.01$, or ***$p \leq 0.001$.

The online version of this article includes the following figure supplement(s) for figure 2:

**Figure supplement 1.** Validation of CFI$_m$25 effects on reporter gene splicing.

**Figure supplement 2.** CFI$_m$25, METTL16, and methionine levels do not regulate MAT2A APA in 293-ATOA cells.

(NHEJ). The latter allele replaces the C-terminal 39 amino acids with 18 amino acids produced from a frameshift. As expected, no MAT2A-DI isoform was detected by northern blot, but methionine depletion leads to an increase of MAT2A mRNA, presumably due to hp2-6-mediated stabilization of the mRNA (*Figure 3B*). Additionally, CFI$_m$25 depletion had no effect on the methionine responsiveness of MAT2A mRNA in 116-ΔDI cells, consistent with a function for CFI$_m$25 in regulation of the MAT2A DI (*Figure 3—figure supplement 1A*). We next compared intracellular SAM levels in 116-Δ DI to the parental line. Similar to 293A-TOA cells (*Figure 2I*), SAM levels decreased upon METTL16 or CFI$_m$25 depletion in HCT116 cells (*Figure 3C*). In marked contrast, these treatments did not decrease SAM levels in 116-ΔDI cells. This observation strongly supports the conclusion that CFI$_m$25 regulates SAM by regulation of the MAT2A DI.

In principle, some of the phenotypes associated with CFI$_m$25 depletion may be due to its regulation of SAM. Since CFI$_m$25 depletion does not alter SAM levels in 116-ΔDI cells, we can use these cells to decouple CFI$_m$25's role in SAM metabolism and APA. To assess APA, we performed Poly(A)-ClickSeq (PAC-seq), a click chemistry technique to sequence the 3′ ends of polyadenylated transcripts (*Elrod et al., 2019*; *Routh, 2019b*; *Routh et al., 2017*). We detected over 2500 poly(A) site changes due to CFI$_m$25 depletion (*Supplementary file 2*). Consistent with previous reports that depletion of CFI$_m$25 favors the use of proximal poly(A) sites, the overwhelming majority of changes were reductions in the percent distal usage (PDU) of poly(A) sites (*Figure 3D*). Importantly, nearly identical levels of shortening were observed in 116-ΔDI (*Figure 3E*). In fact, when we compared the parental and 116-ΔDI lines under either knockdown condition, only 45 and 20 genes experienced APA for the siNT and siCFI$_m$25, respectively (*Figure 3F–G*). Additionally, many exons with significant changes in APA were consistent between HCT116 and 116-ΔDI (*Figure 3H–J*). Because the APA patterns were largely similar even though the SAM levels differ upon CFI$_m$25 knockdown (*Figure 3C*), these global analyses support our conclusion that CFI$_m$25's role in SAM regulation and APA are separable. They also strengthen the conclusion that CFI$_m$25 regulates MAT2A mRNA abundance in a MAT2A DI-dependent fashion.

Consistent with our analysis in 293A-TOA cells (*Figure 2—figure supplement 2*), the PAC-seq analysis showed little use of the weak proximal MAT2A poly(A) site (*Figure 3—figure supplement 1B* and *Supplementary file 2*). Surprisingly, there was a statistically significant increase in use of this site in 116-ΔDI cells upon CFI$_m$25 knockdown, although the distal poly(A) site is still the predominant isoform (*Figure 3—figure supplement 1B*). In addition, there was a novel, albeit low abundance, poly(A) site in intron 7 in the 116-ΔDI cells only. We confirmed the existence of these RNAs by northern blot (*Figure 3—figure supplement 1C–E*). In principle, this could contribute to the restoration of MAT2A mRNA and protein because use of this site excludes the regulatory hp2-6. However, the low levels of each of these isoforms and their lack of induction in 293A-TOA cells (*Figure 2—figure supplement 2A* and *Figure 3—figure supplement 1*) suggest they are not major contributors to MAT2A regulation in our experimental conditions. Thus, while these data may point to a more

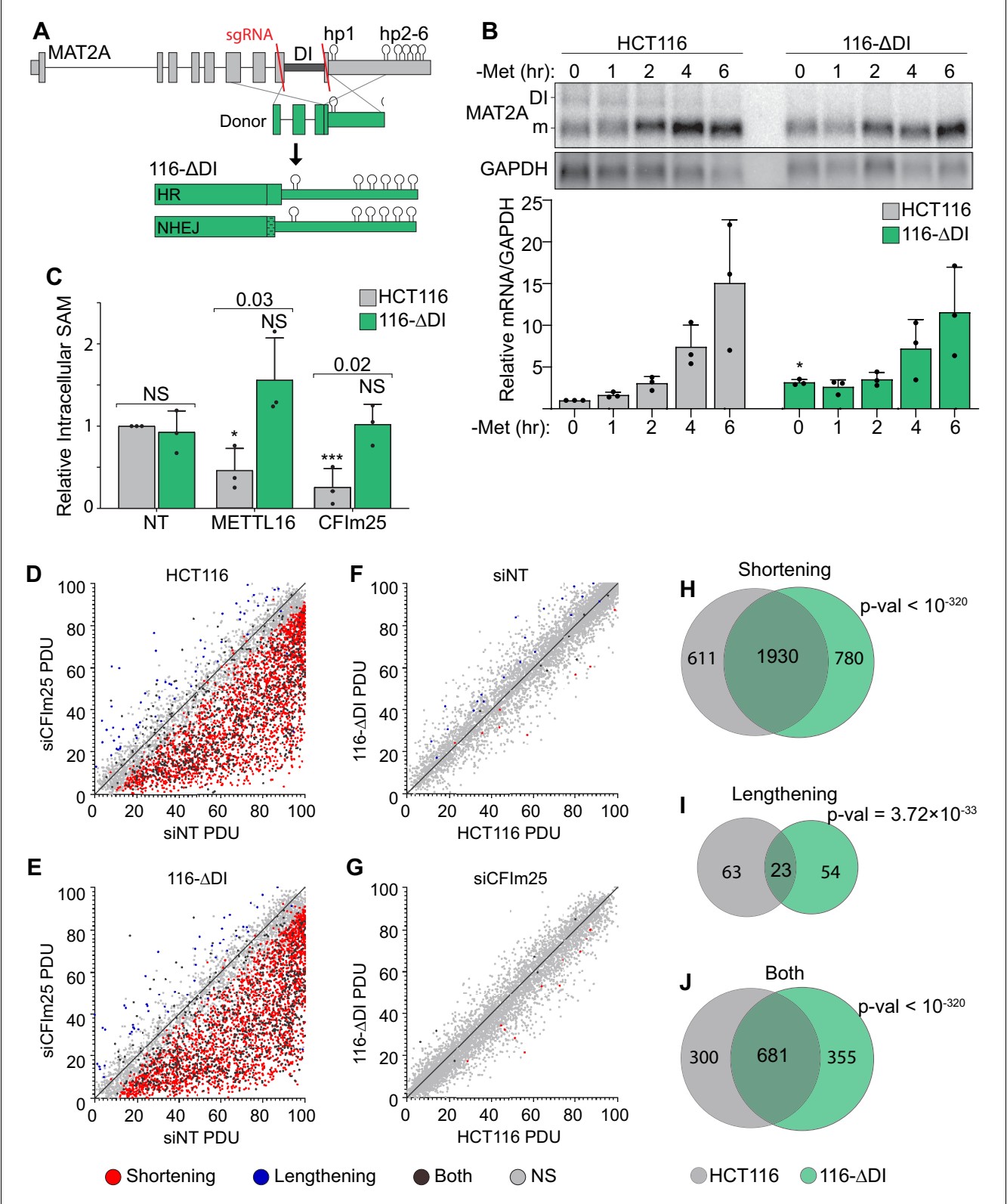

**Figure 3.** CFI<sub>m</sub>25's roles in APA and SAM regulation are separable. (**A**) Schematic of the HCT116 ΔDI (116-ΔDI) cell line. Endogenous MAT2A gene was cut with Cas9 and two sgRNAs (red lines) and repaired with an HR donor plasmid lacking the DI. However, one allele was the result of NHEJ. (**B**) Northern analysis and quantification of MAT2A expression in the HCT116 parental and 116-ΔDI cell lines after the stated methionine depletion times. Samples were normalized to GAPDH and values are relative to HCT116 at 0 hr. Data are mean ± SD; n = 3. Statistics compare 116-ΔDI cells to HCT116

*Figure 3 continued on next page*

*Figure 3 continued*

parental line. (C) Intracellular SAM levels of HCT116 and 116-ΔDI cell lines after a 4-day knockdown with the indicated siRNAs. All values are relative to HCT116 parental non-targeting control. Two statistical comparisons are shown. Significance relative to the matched cell type non-targeting control is annotated with asterisks or NS. The p-values listed above the bars compare the two cell types within each knock-down condition. D-E. APA patterns in HCT116 parental and 116-ΔDI upon CFI$_m$25 depletion. siCFI$_m$25 vs siNT for HCT116 parental (D) and 116-ΔDI (E) plotted by percent distal usage (PDU). Each dot represents a gene with multiple poly(A) clusters, with statistically significant shortening (red), lengthening (blue), or both (dark gray) APA events. Light gray dots are not statistically changed between samples (NS). (F-G) Same as D-E, except HCT116 parental and 116-ΔDI cell lines were compared under siNon-targeting (F) and siCFI$_m$25 (G) conditions. (H-J). Venn diagrams comparing genes with shortening (H), Lengthening (I), or complex APA (Both, J) under CFI$_m$25 depletion for HCT116 (gray) and 116-ΔDI (green) cell lines. p-Values calculated using SuperExactTest (*Wang et al., 2015*).

The online version of this article includes the following figure supplement(s) for figure 3:

**Figure supplement 1.** MAT2A isoforms in HCT116 and 116-ΔDI.

complex regulatory interface between splicing and MAT2A APA, they nonetheless support the conclusion that CFI$_m$25 affects MAT2A processing independent of its well-described roles in APA.

## Two cis-acting CFI$_m$25-binding sites regulate splicing of the MAT2A DI

To further test the role of CFI$_m$25 in MAT2A splicing, we examined publicly available CLIP-seq data to determine if CFI$_m$ interacts with the MAT2A 3′UTR (*Martin et al., 2012*). Because the preferred binding site for CFI$_m$25 is UGUA (*Brown and Gilmartin, 2003*), we initially focused on this motif. All three components of the CFI$_m$ complex cross-linked to varying degrees to the 11 UGUA motifs downstream of hp1 in the MAT2A 3′UTR (*Figure 4A*, ds-UGUA, blue hexagons). To determine if any of these consensus CFI$_m$25-binding sites affect the splicing of MAT2A, we employed a reporter construct consisting of MAT2A exon 8, the DI, and exon 9 with the 3′UTR fused downstream of β-globin coding sequence containing an efficiently spliced β-globin intron (*Figure 4B*; *Pendleton et al., 2017*). Using this construct, we first tested a mutation in the last motif (LM, UGUA to UG$\underline{C}$A), which had the clearest binding in the CLIP-seq data overlapping a UGUA motif (*Figure 4A*). Because this mutation had no effect on intron detention (*Figure 4C–D*, dark blue), we then mutated ten out of eleven sites found in the 3′UTR to $\underline{C}$GUA, U$\underline{C}$UA, or UG$\underline{C}$A (*Figure 4C*, 10/11). We alternated downstream mutations to prevent bias by introducing the same motif multiple times. The tenth site in the 3′UTR was not mutated due to overlap with the conserved METTL16 binding site in hp3 (*Figure 4A* and *Figure 4—figure supplement 1A*; *Doxtader et al., 2018*; *Parker et al., 2011*; *Pendleton et al., 2017*). The 10/11 construct had similar splicing efficiency as wild type, suggesting the downstream UGUA motifs are not involved in MAT2A splicing (*Figure 4C–D*, light blue).

An evolutionarily conserved UGUA motif is found in the DI 9–12 nt upstream of the 3′ splice site, overlapping with a CFI$_m$ complex CLIP-seq peak (*Figure 4A* and *Figure 4—figure supplement 1B*). Due to its proximity to the polypyrimidine tract in the 3′ splice site, we tested two independent point mutations (UG$\underline{C}$A, $\underline{C}$GUA) that maintain pyrimidine content but disrupt the CFI$_m$25 binding motif. We found that both mutations significantly abrogated MAT2A splicing both at basal levels and upon methionine depletion (*Figure 4E–F*). Thus, intronic assembly of the CFI$_m$ complex may contribute to MAT2A splicing.

The strongest and most consistent CLIP-seq peak among all three CFI$_m$ factors lies immediately upstream of hp1 and downstream of the stop codon (*Figure 4A*). There is no UGUA element, but the peak centers on a UGUU motif that is evolutionarily conserved in vertebrates, except in *Danio rerio* where it is a UGUA, the canonical CFI$_m$25 binding site (*Figure 4—figure supplement 1C*). We mutated the UGUU to UG$\underline{C}$U which decreased splicing efficiency comparably to that of the DI point mutations (*Figure 4G–H*, dark red). We observed no changes in splicing efficiency when the UGUU motif was replaced with the canonical UGU$\underline{A}$ sequence (*Figure 4G–H*, pink). Additionally, depletion of METTL16 or CFI$_m$25 resulted in reduced ability to induce splicing of the wild-type UGUU and UGU$\underline{A}$ constructs, while no change was observed for the UG$\underline{C}$U mutation to abrogate splicing (*Figure 4—figure supplement 1D*). Thus, this non-consensus binding site for CFI$_m$ appears to contribute to the splicing of the MAT2A DI.

Consistent with the CLIP-seq and our functional data, structural and biochemical analysis of CFI$_m$ interactions with RNA suggest flexibility in the fourth position of the UGUA motif (*Yang et al., 2011*; *Yang et al., 2010*). To further validate that CFI$_m$ binds to the UGUU site, we conducted label

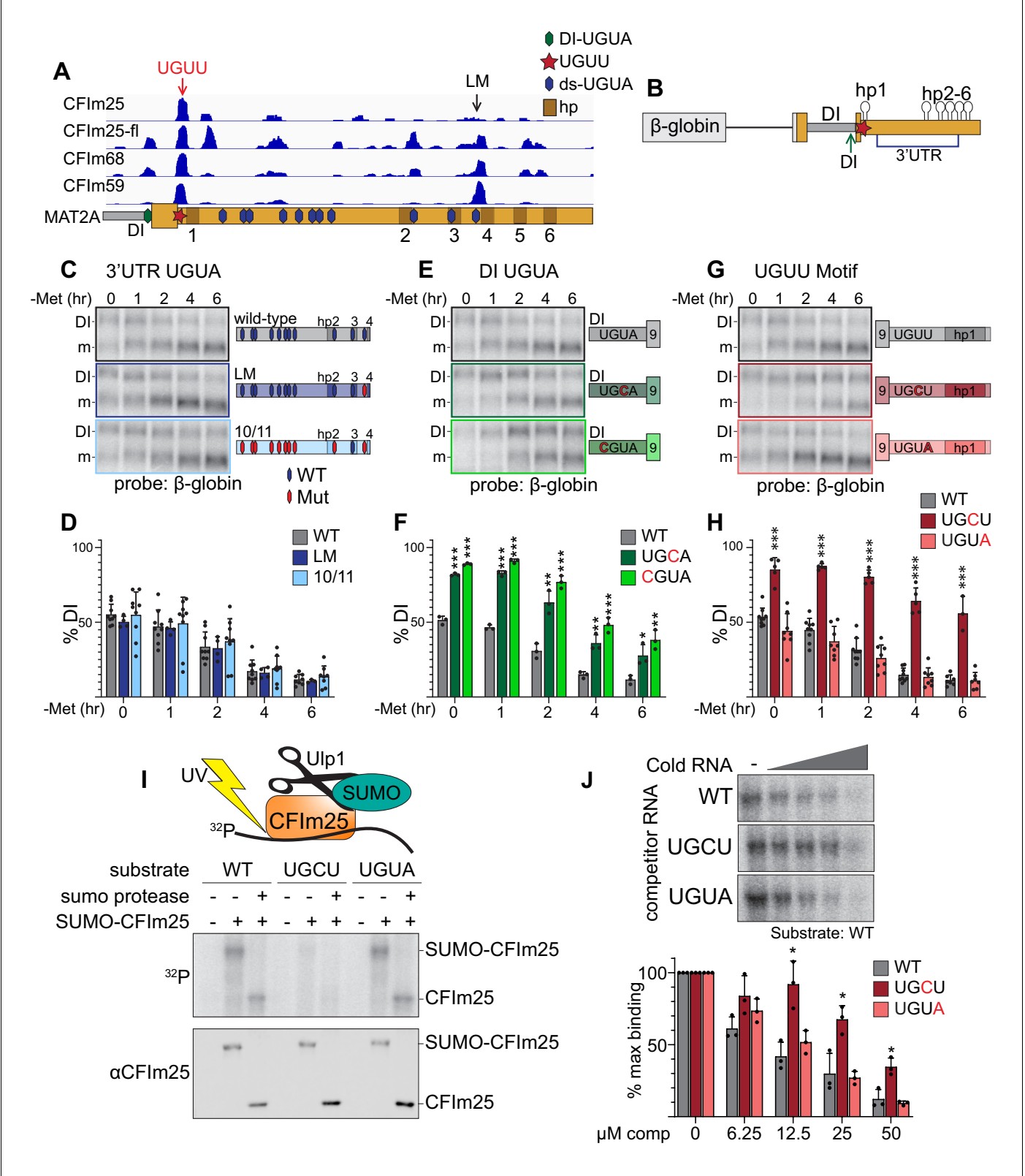

**Figure 4.** CFI$_m$25 binds a non-canonical UGUU motif in MAT2A's 3′UTR. (**A**) IGV browser screen shot of CLIP-seq data of endogenous CFI$_m$25, CFI$_m$68, CFI$_m$59, and flag-tagged CFI$_m$25 (CFI$_m$25-fl) to the MAT2A 3′UTR (*Martin et al., 2012*). Hexagons overlaid on the MAT2A schematic represent UGUA sites (blue, 3′UTR; green, DI UGUA). The UGUU-centered peak is denoted by a red star and red arrow. The last UGUA motif (LM) in the 3′UTR is denoted by a black arrow. MAT2A hairpins are denoted by brown boxes labeled 1–6. (**B**) Schematic of the MAT2A β-globin reporter. The reporter

*Figure 4 continued on next page*

*Figure 4 continued*

consists of a β-globin gene excluding the first intron but maintaining intron two fused to MAT2A exon eight through the end of the 3′UTR. Point mutations were categorized by potential binding site location. Blue bracket denoted 3′UTR includes the 11 downstream UGUA motifs (**C and D**). Green arrow denoted DI, the detained intron UGUA (**E and F**). Red star, the CLIP peak containing a UGUU (**G and H**). (**C and D**) Representative northern blot and quantification of β-globin expression using reporters mutating UGUA motifs in the 3′UTR. In the schematics, hexagons represent UGUA elements, blue hexagons are wild-type, red are mutant. Gray, wild-type reporter (wt). Dark blue, mutation of the LM only (LM). Light blue, 10 of 11 dsUGUA motifs mutated (10/11). n ≥ 3. Note that the representative northern blot data in panels 4C, 4E, and 4G are from the same blot at the same exposure. Wild-type samples were run only once on that gel but are duplicated in the figure for easy formatting and comparison within each group. (**E and F**) Representative northern blot and quantification of β-globin expression using reporters mutating UGUA motif in the MAT2A DI. Gray, wild-type reporter (wt). Green, DI mutants UG<u>C</u>U (dark green) and <u>C</u>GUA (light green). n ≥ 3. (**G and H**) Representative northern blot and quantification of β-globin expression using reporters mutating the UGUU motif immediately upstream of hp1. The UGUU was mutated to UG<u>C</u>U or to the canonical CFI<sub>m</sub> binding motif (UGU<u>A</u>). Gray, wild-type reporter (wt). Dark red, UG<u>C</u>U. Pink, UGU<u>A</u>. n ≥ 3. (**I**) Representative label transfer assay for the CFI<sub>m</sub>25 UGUU-binding motif. SUMO-CFI<sub>m</sub>25 was incubated with radiolabeled 21-nt wild-type substrate centered on the UGUU in the natural sequence; two point-mutants, UG<u>C</u>U and UGU<u>A</u> were also tested. In vitro binding was performed ±SUMO-CFI<sub>m</sub>25 and ±Upl1 SUMO protease as indicated. The top panel is a label transfer (phosphorimager), and the bottom is a western blot showing SUMO-CFI<sub>m</sub>25 loading in each lane. (**J**) Competition label transfer assay. SUMO-CFI<sub>m</sub>25 was incubated with radiolabeled wild-type substrate (UGUU) plus increasing concentrations of cold wild-type or mutant substrate (UG<u>C</u>U, UGU<u>A</u>). Concentrations of competitor RNA increase from left to right (0, 6.25, 12.5, 25, 50 μM). Gray, WT competitor. Red, UG<u>C</u>U competitor. Pink, UGU<u>A</u> competitor. n = 3.

The online version of this article includes the following figure supplement(s) for figure 4:

**Figure supplement 1.** MAT2A contains conserved CFI<sub>m</sub>25-binding sites in the detained intron and 3′UTR.

transfer assays using SUMO-tagged recombinant CFI<sub>m</sub>25. Radiolabeled 21-mer RNA substrates containing the UGUU or its variants were incubated with SUMO-CFI<sub>m</sub>25, cross-linked, then analyzed by SDS-PAGE. We found that CFI<sub>m</sub>25 crosslinked the substrates containing either the canonical UGU<u>A</u> or UGU<u>U</u> motifs more efficiently than to the UG<u>C</u>U substrate (*Figure 4I*). The addition of SUMO protease Upl1 to remove the SUMO tag increased band motility to further confirm that the band represents crosslinked CFI<sub>m</sub>25-RNA complexes. To further test the relative binding of these RNAs to CFI<sub>m</sub>25, we performed a competition assay in which radiolabeled wild-type UGUU substrate was competed with cold UGUU, UG<u>C</u>U, and UGU<u>A</u> RNAs. Cold UG<u>C</u>U RNA was a significantly less efficient competitor than UGUU or UGU<u>A</u> RNAs (*Figure 4J*). Thus, the simplest explanation for the reduced splicing in the UGCU reporters (*Figure 4G–H*) is that CFI<sub>m</sub>25 binds this motif less efficiently than the UGUU or UGU<u>A</u> motifs (*Figure 4I–J*). Together, the CLIP-seq (*Martin et al., 2012*), reporter assays, and in vitro binding all suggest that CFI<sub>m</sub>25 binds to a noncanonical UGUU motif in MAT2A's 3′UTR to regulate intron detention.

## The CFI<sub>m</sub> complex is required for induction of MAT2A splicing

Both CFI<sub>m</sub>25 cofactors CFI<sub>m</sub>68 and CFI<sub>m</sub>59 contain RS domains, so we hypothesized that CFI<sub>m</sub>68 and/or CFI<sub>m</sub>59 promote the splicing of MAT2A. Depletion of CFI<sub>m</sub>59 with two different siRNAs individually or in combination had no significant effects on MAT2A intron detention, while depletion of CFI<sub>m</sub>68 had modest effect for only one siRNA (*Figure 5A–B*; #1). In contrast, co-depletion of CFI<sub>m</sub>68 and CFI<sub>m</sub>59 increased intron detention upon methionine depletion, with a significant increase in intron detention relative to CFI<sub>m</sub>68 knockdown alone (*Figure 5A–B*). Only modest effects of CFI<sub>m</sub>68 and CFI<sub>m</sub>59 depletion were observed in methionine-replete conditions (*Figure 5—figure supplement 1A*). CPSF73 (CPSF3), a CFI<sub>m</sub>-independent component of the cleavage and polyadenylation complex, served as an additional negative control (*Figure 5A–B*; *Chan et al., 2011*). In addition, MAT2A intron detention is nearly identical after exposing cells to media with different methionine concentrations upon CFI<sub>m</sub>68 and CFI<sub>m</sub>59 co-depletion, METTL16, or CFI<sub>m</sub>25 depletion (*Figure 5C–D*). These results suggest possible functional redundancy between CFI<sub>m</sub>68 and CFI<sub>m</sub>59 in the splicing of MAT2A, consistent with the lack of identification of either gene in our CRISPR screen (*Figure 1H*). However, due to co-dependent stability among the members of the CFI<sub>m</sub> complex, variable levels of co-depletion of factors occur (*Figure 5—figure supplement 1B*; *Chu et al., 2019*). Therefore, these data alone do not conclusively show that MAT2A splicing requires CFI<sub>m</sub>68 and CFI<sub>m</sub>59.

To further test a direct role for CFI<sub>m</sub>68 and CFI<sub>m</sub>59 in MAT2A splicing induction, we performed a series of tethering assays. We employed a MAT2A β-globin reporter with two bacteriophage MS2 coat protein-binding sites immediately downstream of hp1. Hp1 in this construct contains an A4G

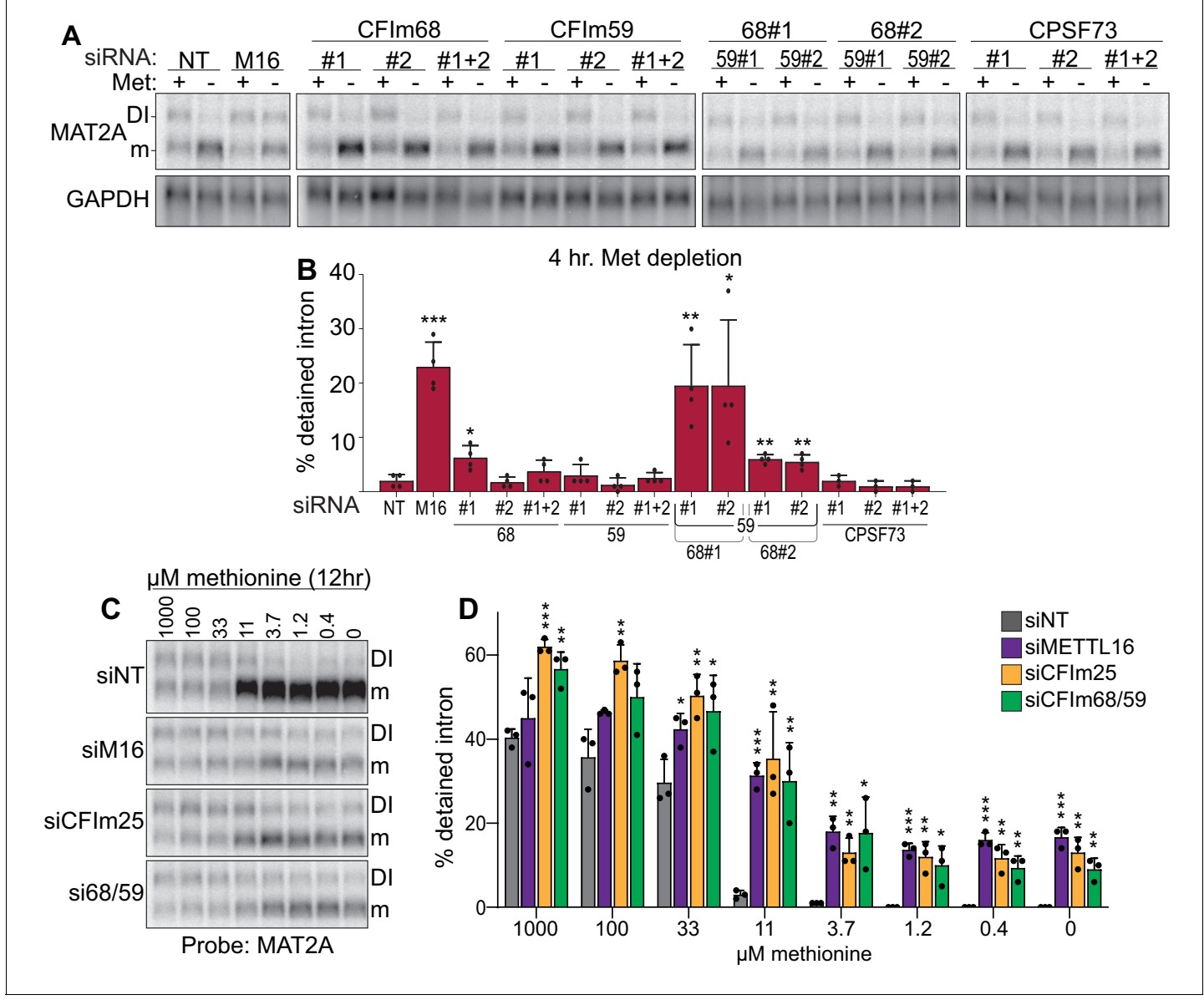

**Figure 5.** Co-depletion of CFI$_m$68 and CFI$_m$59 reduces induction of splicing of the MAT2A DI. (A-B) Representative northern blot and quantification of MAT2A intron detention after CFI$_m$68 and CFI$_m$59 depletion. Two independent siRNAs for each factor were used (labeled #1 and #2). 293A-TOA cells were conditioned with methionine-rich or methionine-free media for 4 hr prior to harvesting. n ≥ 3. (C-D) Representative northern blot and quantification of MAT2A expression after methionine titration after depletion of the indicated factors. 293A-TOA cells were conditioned with media containing the specified methionine concentration for 12 hr prior to harvesting. Non-targeting control, siNT, gray. METTL16, siM16, purple. siCFI$_m$25, orange. siCFI$_m$68 and siCFI$_m$59 co-depletion, green. n = 3.

The online version of this article includes the following figure supplement(s) for figure 5:

**Figure supplement 1.** Additional controls for the CFI$_m$ complex knockdown experiments.

mutation that abrogates METTL16 binding and induction of splicing (*Figure 6A*, asterisk) (*Pendleton et al., 2017*). We co-expressed MS2-coat protein fusions to determine their effects on MAT2A splicing when artificially tethered to the reporter RNA in cells. First, we tethered wild-type (wt) CFI$_m$25 under methionine-rich conditions and observed splicing induction to levels comparable with METTL16 tethering (*Figure 6B*). Importantly, both the MS2-NLS-fl alone and the reporter lacking MS2-binding sites generated low levels of spliced product. We next tested two CFI$_m$25 mutants that bind neither CFI$_m$68 nor CFI$_m$59. The first construct consists of two point-mutations in the nudix

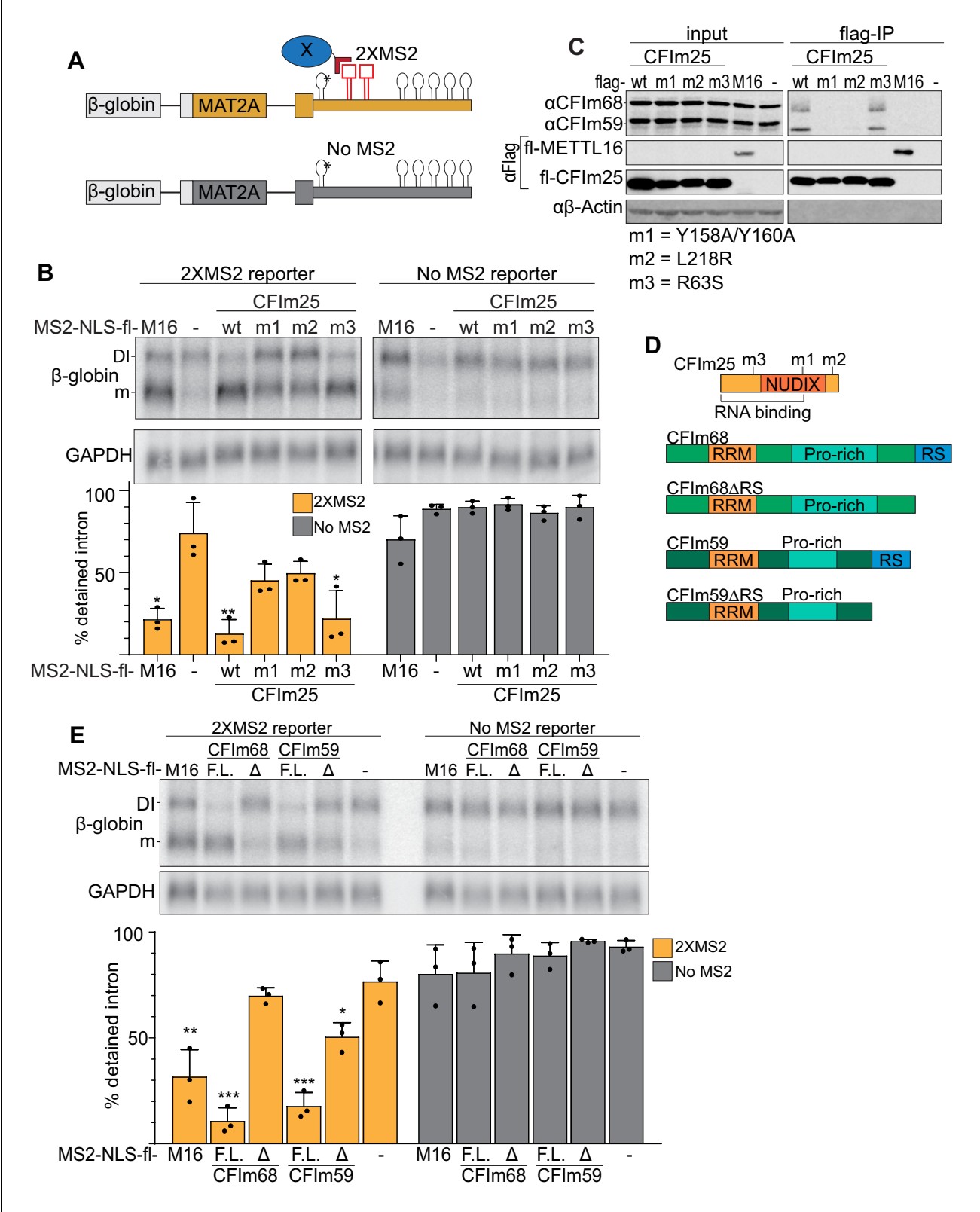

**Figure 6.** Binding of the CFI_m complex is sufficient to promote MAT2A splicing. (**A**) Diagram of the tethering assay. The 2XMS2 β-globin reporter consists of MAT2A exon 8, the detained intron, exon 9, and the full-length 3′UTR with two bacteriophage MS2-coat protein binding sites inserted 3′ of A4G mutant hp1 (asterisk). A matched reporter lacking the MS2 site ('No MS2') is used to measure background. All MS2 fusion proteins have an N-terminal MS2 coat protein, an SV40 nuclear localization signal, and flag tag (MS2-NLS-fl). The MS2-NLS-fl alone is expressed as negative control.

*Figure 6 continued on next page*

*Figure 6 continued*

Diagram not to scale. (B) Northern analysis of β-globin reporter RNA after tethering CFI$_m$25 variants. MS2-NLS-fl fusions to METTL16 (M16), CFI$_m$25 variants (wt, m1, m2, m3), or MS2-NLS-fl alone (-) and were expressed with the indicated reporters. Statistical analysis is relative to matched MS2-vector control. Orange, 2XMS2 reporter. Gray, No MS2 reporter. n = 3. (C) Coimmunoprecipitation of CFI$_m$68 and CFI$_m$59 with flag-CFI$_m$25. Flag-tagged wild-type CFI$_m$25 (wt) or mutants (m1, m2, m3), flag-tagged METTL16 (M16), or flag-vector (-) were expressed in HEK293 cells before immunoprecipitation with anti-flag beads. The immunoprecipitates were then probed for endogenous CFI$_m$68 and CFI$_m$59, flag, or β-actin. Input is 10% of the lysate volume applied to flag beads. n = 3. (D) Diagrams of CFI$_m$25, CFI$_m$68, CFI$_m$68ΔRS, CFI$_m$59, and CFI$_m$59ΔRS proteins; diagrams to scale. (E) Northern analysis of β-globin reporter RNA after tethering CFI$_m$68 or CFI$_m$59. MS2-NLS-flag-tagged METTL16, CFI$_m$68, CFI$_m$68ΔRS, CFI$_m$59, CFI$_m$59ΔRS, or vector (-) were co-transfected with the 2XMS2 β-globin reporter or no MS2 reporter control. Statistical analysis is relative to matched MS2-vector control. F.L., full length; Δ, ΔRS domain. n = 3.

The online version of this article includes the following figure supplement(s) for figure 6:

**Figure supplement 1.** Expression of full-length and RS-deletion MS2-NLS-fl CFI$_m$68 and CFI$_m$59 proteins.

hydrolase domain (m1, Y158A/Y160A) and the second contains a single point mutation near the C-terminus (m2, L218R)(*Figure 6D*; *Yang et al., 2011*; *Zhu et al., 2018*). We confirmed that CFI$_m$25 binding to CFI$_m$68 and CFI$_m$59 was abrogated in both mutants by coimmunoprecipitation (*Figure 6C*). Upon tethering of these CFI$_m$ mutants, we observed a significant loss in splicing induction suggesting that a functional CFI$_m$ complex is responsible for splicing (*Figure 6B*). Importantly, the CFI$_m$25 mutants expressed to comparable levels to that of the wild-type construct (*Figure 6C*, input). In contrast, tethering of an RNA-binding mutant (m3, R63S) of CFI$_m$25 maintained splicing activity, as expected since RNA binding is driven by MS2 tethering (*Yang et al., 2011*; *Yang et al., 2010*).

CFI$_m$25 has no known splicing domains, but CFI$_m$68 and CFI$_m$59 both contain N-terminal RS domains. Therefore, we reasoned that the RS domains of CFI$_m$68 and CFI$_m$59 may be responsible for the splicing of MAT2A. If so, tethering full-length CFI$_m$68/CFI$_m$59 will induce splicing of MAT2A, while the tethering of CFI$_m$68/CFI$_m$59 lacking the RS domains (ΔRS) will not (*Figure 6D*). As predicted, CFI$_m$68 and CFI$_m$59 tethering was sufficient to induce splicing, while the ΔRS proteins were unable to affect intron detention (*Figure 6E*). Lack of splicing was not due to inability of the constructs to express, as the RS domain deletion expressed to comparable levels of their full-length counterparts (*Figure 6—figure supplement 1*). Together, these observations suggest that the CFI$_m$ complex is responsible for the splicing of the MAT2A DI.

## CFI$_m$ is downstream of METTL16 in the MAT2A splicing induction mechanism

Our data suggest the model for CFI$_m$ complex-mediated induction of MAT2A splicing shown in *Figure 7A*. CFI$_m$ is recruited to the UGUA in the DI and UGUU in MAT2A 3′UTR. Upon CFI$_m$ binding, CFI$_m$68 and CFI$_m$59 serve as the downstream effectors that promote efficient splicing of the DI through their RS domains. This model predicts that CFI$_m$ is downstream of the SAM sensor METTL16. To test the proposed hierarchy of factors in the splicing of the MAT2A DI, we combined our tethering system with siRNA knockdown of individual factors. In some cases, the expression of the MS2 transgenes is reduced upon knockdown of these essential factors (*Figure 7—figure supplement 1*). Importantly, in each of these cases, our data further show that the lower levels remain sufficient to potently drive reporter splicing (see *Figure 7—figure supplement 1* legend). If CFI$_m$ is downstream of METTL16, tethering METTL16 will no longer induce splicing upon CFI$_m$25 depletion. Indeed, the tethering of METTL16 fails to induce splicing after CFI$_m$25 depletion compared to the non-target and no MS2 controls (*Figure 7B*, gray and orange). Conversely, tethering of CFI$_m$25, so long as it is in complex with CFI$_m$68 or CFI$_m$59, should induce splicing even in METTL16-depleted cells. As expected, tethering of CFI$_m$25 after METTL16 depletion results in splicing induction comparable to the non-targeting control (*Figure 7B*, gray and purple). This supports the idea that METTL16 acts as the upstream SAM sensor while CFI$_m$ directly mediates the splicing of MAT2A.

Our model proposes that CFI$_m$68 or CFI$_m$59 interchangeably function for the splicing of MAT2A, while CFI$_m$25 recruits CFI$_m$68/CFI$_m$59 to the RNA. If so, tethering METTL16 or CFI$_m$25 will no longer induce splicing if the downstream effectors CFI$_m$68 and CFI$_m$59 are not present. Consistent with this prediction, CFI$_m$68/CFI$_m$59 co-depletion reduces splicing induction upon METTL16 or CFI$_m$25 tethering (*Figure 7C*, green). Surprisingly, depletion of CFI$_m$68 alone had similar effects as CFI$_m$68/CFI$_m$59

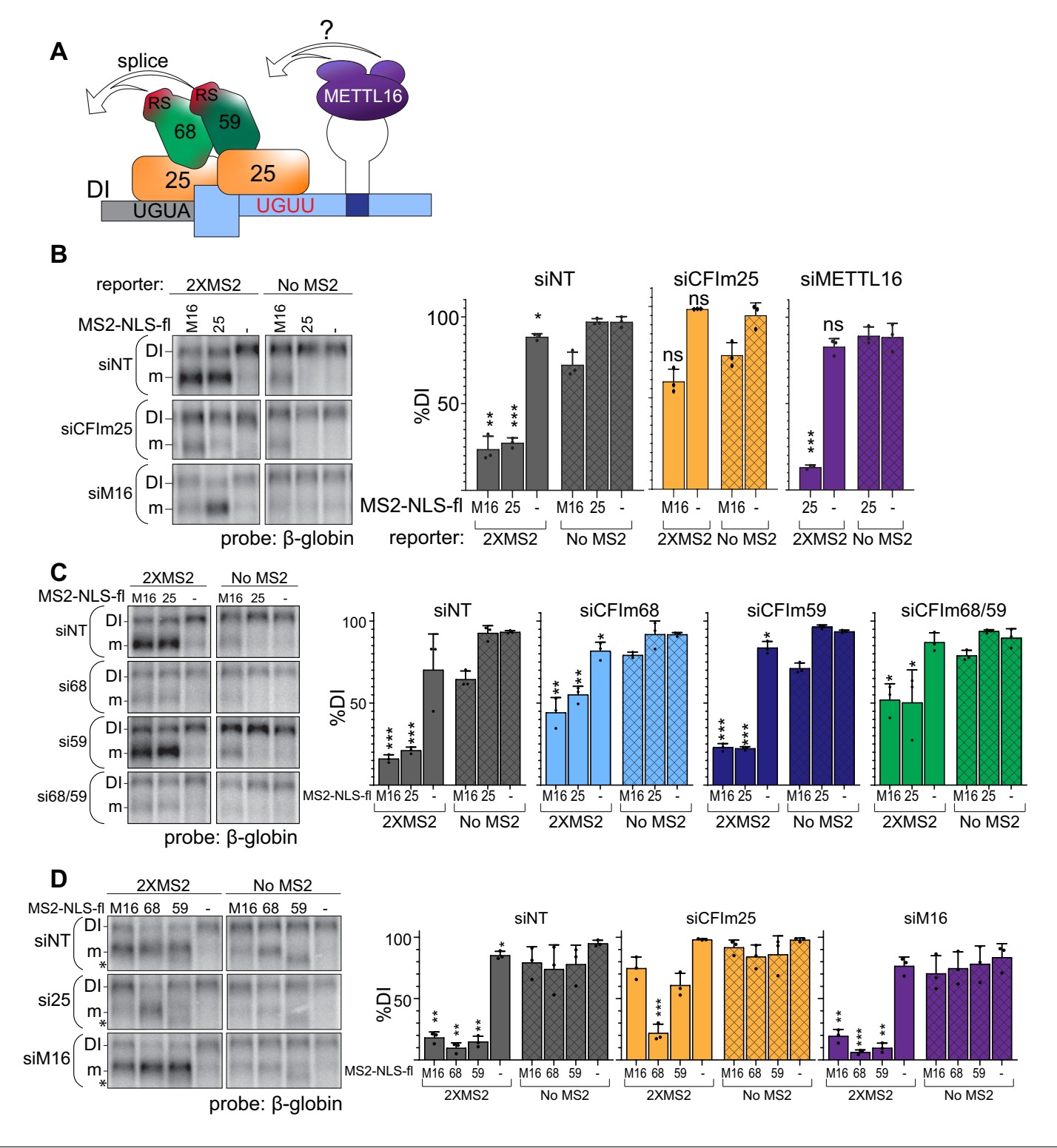

**Figure 7.** CFI$_m$ is the downstream splicing effector of METTL16. (**A**) Working model for CFI$_m$-induced MAT2A splicing. The CFI$_m$ complex binds to a non-canonical UGUU motif in the MAT2A 3′UTR and UGUA in the detained intron. The model depicts a single complex binding both sites through CFI$_m$25 dimers, but our data are equally consistent with independent binding of CFI$_m$ complexes to each site (see Discussion). In either case, we propose CFI$_m$ binding promotes MAT2A splicing by proximity of the RS domains in CFI$_m$68 or CFI$_m$59. How this is integrated with METTL16 binding remains unknown. For simplicity, this model focuses solely on splicing activity and does not depict the contributions of METTL16 and hp2-6 in MAT2A RNA stability. (**B**) Northern analysis of β-globin reporter RNA after knockdown of the indicated factor and tethering of MS2-NLS-fl tagged METTL16

*Figure 7 continued on next page*

*Figure 7 continued*

(M16), CFI$_m$25 RNA binding mutant (m3, 25), or MS2-NLS-fl vector (-). *Left*, representative northern blot. *Right*, quantification by percent detained intron for non-targeting siRNA (siNT, gray), or siRNAs targeting CFI$_m$25 (si25, orange) or METTL16 (siM16, purple). n = 3. (C) Same as (B). Non-targeting siRNA (siNT, gray), siCFI$_m$68 (si68, light blue), siCFI$_m$59 (si59, dark blue), and siCFI$_m$68/59 co-depletion (si68/59, green). n = 3. (D) Same as (B) except tethering of MS2-NLS-fl-METTL16 (M16), -CFI$_m$68 (68), -CFI$_m$59 (59), or MS2-NLS-fl vector (-) to the MS2 reporter constructs in cells depleted of METTL16 or CFI$_m$25. Overexpression of CFI$_m$59 caused a band of unknown identity to appear (asterisk). The band was cell-type specific: it did not appear in HEK293 cells (*Figure 6E*), but it does in 293A-TOA cells used here. Non-targeting siRNA (siNT, gray), siCFI$_m$25 (si25, orange), and siMETTL16 (siM16, purple). n = 3. Statistical analyses in (A, B, C, D) compare 2XMS2 reporter to matched, no-MS2 reporter.

The online version of this article includes the following figure supplement(s) for figure 7:

**Figure supplement 1.** Expression of MS2-NLS-fl proteins after depletion of METTL16 or CFI$_m$ components.

co-depletion, while CFI$_m$59 depletion was comparable to the non-targeting control. We likewise observed that tethering CFI$_m$68 induces splicing in CFI$_m$25-depleted cells, but CFI$_m$59 did not (*Figure 7D*, orange). These data demonstrate that tethering CFI$_m$68 is sufficient to induce splicing of MAT2A in the absence of METTL16 or other CFI$_m$ complex family members. The function of CFI$_m$59 in MAT2A splicing is less clear, but it may contribute to splicing redundantly with CFI$_m$68 or have other functions in MAT2A splicing or CFI$_m$ complex assembly (see Discussion).

## Discussion

Here, we propose that the CFI$_m$ complex, a major APA factor, is a key regulator of SAM metabolism. Next to METTL16, CFI$_m$25 was a top candidate in a global unbiased screen for regulators of MAT2A splicing. Knockdown and tethering independently support the conclusion that components of the CFI$_m$ complex drive splicing of the MAT2A DI through the RS domains of CFI$_m$68 and/or CFI$_m$59. Moreover, mutation of CFI$_m$25 binding sites in the MAT2A DI and 3′UTR both abrogate splicing. Taken together, these data support our conclusion that the CFI$_m$ complex is a downstream effector that promotes MAT2A splicing regulated by the SAM sensor METTL16 (*Figure 7A*).

Our data suggest that MAT2A splicing regulation is independent of its role in 3′ end formation. Whether APA-independent regulation of splicing by CFI$_m$ is unique to MAT2A or is more widespread remains unknown. However, CFI$_m$25 has recently been reported to affect regulation of the glutaminase (GLS) gene in a complex regulatory paradigm that includes alternative splicing and APA (*Masamha et al., 2014*; *Redis et al., 2016*). GLS generates at least three isoforms, including two longer transcripts that use the same last exon but distinct polyadenylation sites within that exon. A shorter isoform, called GAC, uses a unique 3′ splice site and polyadenylation site within an upstream intron to generate an alternative last exon. CFI$_m$25 promotes the accumulation of the GAC isoform, suggesting it defines this alternative last exon by promoting usage of its poly(A) site and/or the 3′ splice site. Interestingly, the 3′ splice site of the GAC isoform has a weak polypyrimidine tract and includes a UGUA within 20 nt of the 3′ AG splice site, similar to MAT2A. Thus, other genes may require CFI$_m$ binding near a 3′ splice site to promote splicing, but the extent of the phenomenon is unclear.

Our proposed model leads to an important question: how is METTL16 binding to hp1 connected to CFI$_m$-mediated MAT2A DI splicing? The simplest model is that METTL16 recruits CFI$_m$ to the RNA, but we were unsuccessful in coimmunoprecipitating METTL16 and CFI$_m$25. Additionally, our unpublished yeast two-hybrid experiments and coimmunoprecipitation mass spectrometry experiments were consistent with a published report that found no METTL16-binding proteins (*Ignatova et al., 2019*). It seems unlikely that METTL16 functions without other binding partners, but if METTL16 and CFI$_m$25 directly interact, this interaction is likely transient. Alternatively, METTL16 may mediate CFI$_m$25 binding to MAT2A through an RNA conformational change that exposes the binding sites. This conformational change hypothesis is supported by the discovery that the VCR domain of METTL16 binds RNA (*Aoyama et al., 2020*). Since the VCR domains alone can drive splicing when tethered to a reporter (*Pendleton et al., 2017*), the VCR domain could alter RNA secondary structure surrounding hp1 thereby allowing CFI$_m$25 binding. Alternatively, other RNA, proteins, or posttranslational modifications may promote the transient formation of the METTL16:CFI$_m$ complex on the MAT2A transcript. Such proteins would likely function redundantly with other factors because they were not identified in our CRISPR screen. In any case, future

experiments will focus on distinguishing among these and other models that explain the interface between METTL16 and CFI$_m$ in MAT2A splicing.

CFI$_m$-binding sites in the MAT2A DI and immediately upstream of hp1 are necessary for MAT2A splicing. Mutation of either the DI UGUA or 3′UTR UGUU results in abrogation of splicing to comparable levels, suggesting that both sites contribute equally to the splicing of MAT2A (*Figure 4E–H*). The two sites could independently function as CFI$_m$ binding sites. Alternatively, since CFI$_m$25 exists as dimer within the complex, it is possible that the sites are bound simultaneously with the intervening RNA looped out. In fact, previous data suggests looping may be important for the function of CFI$_m$25 (*Yang et al., 2011*; *Yang et al., 2010*). Two UGUA motifs in the PAPOLA mRNA 3′UTR simultaneously bind to the CFI$_m$25 dimer to promote more efficient binding. The two sites must be at least nine nt apart to allow for looping, but the upper limit of distance between UGUA motifs has yet to be determined. The MAT2A DI UGUA and UGUU motif upstream of hp1 are 114 nt apart allowing ample separation for RNA looping. It has been proposed that RNA looping mediated by the CFI$_m$ complex regulates poly(A) site selection by occluding or presenting poly(A) sites (*Yang et al., 2011*). However, to our knowledge, this has yet to be demonstrated in cells. Given that one of the sites is in the DI near the 3′ splice site (*Figure 4—figure supplement 1B*), it is tempting to speculate that CFI$_m$-dependent RNA looping exposes the 3′ splice site to increase its accessibility to the spliceosome.

In addition to its potential role in RNA looping, the UGUA in the DI presents a particularly interesting component of MAT2A splicing regulation. The 3′ splice site of most pre-mRNAs is recognized by binding of the U2AF heterodimer that binds the 3′ AG and the ~9 nt polypyrimidine tract through the U2AF1 (U2AF35) and U2AF2 (U2AF65) subunits, respectively. After binding the 3′ splice site, U2AF recruits the spliceosomal U2 snRNP (*Lee and Rio, 2015*; *Ruskin et al., 1988*; *Sibley et al., 2016*; *Wu and Fu, 2015*). The polypyrimidine tract in the MAT2A DI is interrupted by three purines, two of which are the part of the UGUA (*Figure 4—figure supplement 1B*). Thus, our data suggest that the same sequence that weakens the polypyrimidine tract also provides a regulatory CFI$_m$ binding site capable of overcoming inefficient splicing.

We can imagine at least three distinct possibilities for how CFI$_m$ promotes splicing within the MAT2A detained intron. In the first model, CFI$_m$ recruits U2AF such that it binds directly to the weak 3′ splice site. Upon recruitment, CFI$_m$ may 'handover' the RNA to U2AF, as has been described for some RNA export factors (*Hautbergue et al., 2008*). Alternatively, it is possible that CFI$_m$ recruits U2AF, but only one of the U2AF RNA recognition motifs (RRMs) engages the RNA at the four pyrimidines immediately adjacent to the AG (*Figure 4—figure supplement 1B*). Consistent with this idea, U2AF's RRMs are separated by a flexible linker and each interacts with ~4–5 nt (*Agrawal et al., 2016*; *Jenkins et al., 2013*). A related second model proposes that CFI$_m$ recruits U2AF, but U2AF does not directly bind the RNA. Instead, U2AF functions exclusively by protein-protein interactions within the CFI$_m$-RNA complex. These recruitment models are similar to the mechanism in which the YB-1 protein binds RNA to recruit U2AF to weak splice sites (*Wei et al., 2012*). Moreover, protein-protein interactions between CFI$_m$ and U2AF2 have been demonstrated (*Millevoi et al., 2006*). However, we note that those interactions are specific for CFI$_m$59, which is not strictly required to activate MAT2A splicing. A third model proposes that CFI$_m$ functionally substitutes for U2AF. In fact, as many as ~12% of introns may be U2AF-independent (*Shao et al., 2014*). In addition, U2AF can be functionally substituted by SPF45/RBM17 on a subset of introns (*Fukumura et al., 2020*). Moreover, in vitro U2AF depletion can be complemented by excess SR proteins (*MacMillan et al., 1997*), so it is not unreasonable to suggest that the RS domains of CFI$_m$ may supply this function for MAT2A DI.

We have determined that the CFI$_m$ complex is necessary for the splicing of MAT2A, but the composition of this complex is unclear. In our knockdown experiments CFI$_m$68 and CFI$_m$59 appear to be redundant for splicing (*Figure 5*) and neither appeared as CRISPR screen hits (*Figure 1* and *Supplementary file 1*). Further supporting functional redundancy, tethering of either CFI$_m$68 or CFI$_m$59 resulted in comparable splicing induction, as long as an RS domain is present (*Figure 6E*). Despite CFI$_m$68 and CFI$_m$59 belonging to the same complex and sharing significant sequence and structural similarity, CFI$_m$68 and CFI$_m$59 have non-redundant function in cells (*Deng et al., 2019*; *Li et al., 2011*; *Tresaugues et al., 2010*; *Yang et al., 2011*). Knockdown of CFI$_m$68 leads to a global shift to proximal poly(A) site usage, similarly to CFI$_m$25 knockdown, while CFI$_m$59 has no effect (*Gruber et al., 2012*; *Zhu et al., 2018*). Additionally, CFI$_m$68 and CFI$_m$59 have distinct interaction partners (*Martin et al., 2012*; *Martin et al., 2010*; *Millevoi et al., 2006*). Although CFI$_m$68 and

CFI$_m$59 initially appear to be functionally similar for the splicing of MAT2A, some of our data minimize the role of CFI$_m$59. When tethered, CFI$_m$25 is incapable of promoting splicing in the absence of CFI$_m$68, while CFI$_m$59 knockdown alone has no effect (*Figure 7C*). Additionally, tethering of CFI$_m$68 enables splicing independent of CFI$_m$25, while CFI$_m$59 requires CFI$_m$25 (*Figure 7D*). One possible explanation is rooted in the three possible compositions of the CFI$_m$ complex. The CFI$_m$25 dimer can form a tetramer with two CFI$_m$59 proteins, two CFI$_m$68 proteins, or with one of each. Thus, tethering of CFI$_m$59 may lead to recruitment of the CFI$_m$ complex that includes CFI$_m$68. If so, then tethering of a CFI$_m$25:CFI$_m$68:CFI$_m$59 containing complex would be capable of promoting splicing but would require CFI$_m$25 to recruit CFI$_m$68. More testing needs to be done to unravel the overlapping and distinct roles of CFI$_m$68 and CFI$_m$59 in regulation of MAT2A.

The CFI$_m$ complex is a major regulator of poly(A) site selection (*Brumbaugh et al., 2018*; *Gruber et al., 2012*; *Li et al., 2015*; *Martin et al., 2012*; *Masamha et al., 2014*; *Zhu et al., 2018*). Our data show that the CFI$_m$ complex's role in MAT2A splicing and SAM metabolism appear to be independent of poly(A) site selection (*Figure 3D–G*). The shortening of 3′UTRs upon CFI$_m$ knockdown has been linked to important biological phenotypes including cancer cell proliferation and cell differentiation (*Brumbaugh et al., 2018*; *Chu et al., 2019*; *Jafari Najaf Abadi et al., 2019*; *Masamha et al., 2014*; *Tamaddon et al., 2020*). For example, knockdown of either CFI$_m$25 or CFI$_m$68 increases transcription-factor-induced reprogramming of mouse embryonic fibroblasts (MEFs) into induced pluripotent stem cells (iPSC)(*Brumbaugh et al., 2018*; *Ji and Tian, 2009*). This effect is attributed to global shifts from distal poly(A) site usage to proximal, which is characteristic of less differentiated and more proliferative cells (*Ji et al., 2009*; *Mayr and Bartel, 2009*; *Sandberg et al., 2008*; *Shepard et al., 2011*). Our work suggests that SAM levels may drop in MEFs upon CFI$_m$ knockdown, so it is possible that this SAM reduction contributes to the increased dedifferentiation potential, priming cells for transition into iPSC (*Shiraki et al., 2014*). Similarly, it is plausible that SAM levels contribute to the growth potential of cancer cells, as cancer cells often have reduced methylation potential and/or intracellular SAM levels (*Gao et al., 2019*; *Murray et al., 2019*). Therefore, CFI$_m$ knockdown may augment cancer growth by reducing SAM levels in addition to the well-defined 3′UTR shortening mechanism (*Jafari Najaf Abadi et al., 2019*). Thus, it will be interesting to decouple CFI$_m$25's role in poly(A) site selection from that in maintenance of intracellular SAM to see how each contributes to these important biological phenomena.

## Materials and methods

**Key resources table**

| Reagent type (species) or resource | Designation | Source or reference | Identifiers | Additional information |
| --- | --- | --- | --- | --- |
| Strain, strain background (*E. coli*) | Bacterial: ElectroMAX Stbl4 Competent Cells | Thermo Fisher | Cat#: 11635018 | Electrocompetent Cells |
| Strain, strain background (*E. coli*) | Bacterial: Rosetta (DE3) cells | EMD Millipore | Cat#: 70954 | For production of SUMO-CFI$_m$25 |
| Cell line (*H. sapiens*) | HEK293 | Dr. Joan A. Steitz (Yale University) | *Conrad and Steitz, 2005* | |
| Cell line (*H. sapiens*) | HEK293T | Dr. Joshua Mendell | UT Southwestern Medical Center | |
| Cell line (*H. sapiens*) | 293A-TOA | Dr. Nicholas K. Conrad | UT Southwestern Medical Center (*Sahin et al., 2010*) | |
| Cell line (*H. sapiens*) | HCT116 | ATCC | CCL-247 | |
| Cell line (*H. sapiens*) | HCT116-GFP-β-MAT8-3′ | This paper | 'Reporter' | Maintained by Nicholas K. Conrad lab |
| Cell line (*H. sapiens*) | HCT116-ΔDetainedIntron (116-ΔDI) | This paper | '116-ΔDI' | Maintained by Nicholas K. Conrad lab |

*Continued on next page*

*Continued*

| Reagent type (species) or resource | Designation | Source or reference | Identifiers | Additional information |
|---|---|---|---|---|
| Cell line (*H. sapiens*) | HCT116-GFP-T2A-β-MAT8-3′-hp2-6m9#1 | This paper | 'T2A, hp2-6m9#1' | Maintained by Nicholas K. Conrad lab |
| Cell line (*H. sapiens*) | HCT116-GFP-T2A-β-MAT8-3′-hp2-6m9#2 | This paper | 'T2A, hp2-6m9#2' | Maintained by Nicholas K. Conrad lab |
| Antibody | Rabbit polyclonal anti-GFP | Abcam | Cat#: ab6556; RRID:AB_305564 | (1:2000) |
| Antibody | Rabbit polyclonal anti-MAT2A | Novus | Cat#: NB110-94158; RRID:AB_1237164 | (1:2000) |
| Antibody | Mous monoclonal anti-β-actin | Abcam | Cat# ab6276; RRID:AB_2223210 | (1:5000) |
| Antibody | Rabbit polyclonal anti-METTL16 | Bethyl | Cat# A304-192A; RRID:AB_2620389 | (1:5000) |
| Antibody | Rabbit monoclonal anti-CFIm25 | Invitrogen | Cat#: 702871; RRID:AB_2723420 | (1:2000) |
| Antibody | Rabbit polyclonal anti-CFIm59 | Bethyl | Cat# A301-360A; RRID:AB_937864 | (1:2000) |
| Antibody | Rabbit polyclonal anti-CFIm68 | Bethyl | Cat#: A301-358A; RRID:AB_937785 | (1:2000) |
| Antibody | Rabbit polyclonal anti-flag | Sigma | Cat#: F7425; RRID:AB_439687 | (1:5000) |
| Antibody | Mouse monoclonal anti-Bacteriophage MS2 Coat Protein | Kerafast | Cat#: ED0005 | (1:2000) |
| Antibody | Mouse monoclonal anti-flag | Sigma | Cat#: F3165; RRID:AB_259529 | (1:2000) |
| Antibody | Goat anti-mouse | IRDye | Cat#:926–68020; RRID:AB_10706161 | (1:10,000) |
| Antibody | Goat anti-mouse | IRDye | Cat#: 926–32210; RRID:AB_621842 | (1:10,000) |
| Antibody | Goat anti-rabbit | IRDye | Cat#: 926–32211; RRID:AB_621843 | (1:10,000) |
| Recombinant DNA reagent | Plasmid: pcDNA-flag | *Sahin et al., 2010* | N/A | |
| Recombinant DNA reagent | Plasmid: pcMS2-NLS-flag | *Sahin et al., 2010* | N/A | |
| Recombinant DNA reagent | Plasmid: β-MAT-WT | *Pendleton et al., 2017* | 'WT' | |
| Recombinant DNA reagent | Plasmid: β-MAT-preHP1-UGCU | This paper | 'UGCU' | Contains UGUU upstream of hp1 to UG**C**U mutation in β-MAT-WT backbone. See details in 'Plasmids' section of 'Materials and Methods' |
| Recombinant DNA reagent | Plasmid: β-MAT-preHP1-UGUA | This paper | 'UGUA' | Contains UGUU upstream of hp1 to UGU**A** mutation in β-MAT-WT backbone. See details in 'Plasmids' section of 'Materials and Methods' |
| Recombinant DNA reagent | Plasmid: β-MAT-3′ UTR-Last-Motif-UGCA | This paper | 'LM' | Contains UGUA LM 3′UTR to UG**C**A mutation in β-MAT-WT backbone. See details in 'Plasmids' section of 'Materials and Methods' |

*Continued on next page*

*Continued*

| Reagent type (species) or resource | Designation | Source or reference | Identifiers | Additional information |
|---|---|---|---|---|
| Recombinant DNA reagent | Plasmid: β-MAT-3′ UTR-10/11 | This paper | '10/11' | Contains point mutations in 10/11 UGUA motifs of 3'UTR in β-MAT-WT backbone. See details in 'Plasmids' section of 'Materials and Methods' |
| Recombinant DNA reagent | Plasmid: β-MAT-DI-UGCA | This paper (Materials and Methods) | 'UGCA' | Contains detained intron UGUA to UG**C**A mutation in β-MAT-WT backbone. See details in 'Plasmids' section of 'Materials and Methods' |
| Recombinant DNA reagent | Plasmid: β-MAT-DI-CGUA | This paper | '**C**GUA' | Contains detained intron UGUA to **C**GUA mutation in β-MAT-WT backbone. See details in 'Plasmids' section of 'Materials and Methods' |
| Recombinant DNA reagent | Plasmid: β-MAT-hp1G4 | *Pendleton et al., 2017* | 'No MS2' | |
| Recombinant DNA reagent | Plasmid: β-MAT-hp1G4, 2XMS2 | This paper | '2XMS2' | Contains two MS2 binding sites inserted immediately downstream of hp1. See details in 'Plasmids' section of 'Materials and Methods' |
| Recombinant DNA reagent | Plasmid: pcMS2-NLS-Flag-METTL16 | *Pendleton et al., 2017* | 'MS2-NLS-fl-M16' | |
| Recombinant DNA reagent | Plasmid: Flag-METTL16 | *Pendleton et al., 2017* | 'M16' | |
| Recombinant DNA reagent | Plasmid: pcMS2-NLS-Flag-CFI$_m$25 | This paper | 'wt' | CFIm25 inserted into pcMS2-NLS-Flag vector. See details in 'Plasmids' section of 'Materials and Methods' |
| Recombinant DNA reagent | Plasmid: pcMS2-NLS-Flag-CFI$_m$25-m1 (Y158A/Y160A) | This paper | 'm1' | Y158A/Y160A mutations made in pcMS2-NLS-Flag-CFI$_m$25. See details in 'Plasmids' section of 'Materials and Methods' |
| Recombinant DNA reagent | Plasmid: pcMS2-NLS-Flag-CFI$_m$25-m2 (L218R) | This paper | 'm2' | L218R mutation made in pcMS2-NLS-Flag-CFI$_m$25. See details in 'Plasmids' section of 'Materials and Methods' |
| Recombinant DNA reagent | Plasmid: pcMS2-NLS-Flag-CFI$_m$25-m3 (R63S) | This paper | 'm3'; 'MS2-NLS-fl-25' | R63S mutation made in pcMS2-NLS-Flag-CFI$_m$25. See details in 'Plasmids' section of 'Materials and Methods' |
| Recombinant DNA reagent | Plasmid: pcMS2-NLS-Flag-CFI$_m$68 | This paper | 'CFIm68 F.L."', '68', 'MS2-NLS-fl-68' | CFIm68 inserted into pcMS2-NLS-Flag vector. See details in 'Plasmids' section of 'Materials and Methods' |
| Recombinant DNA reagrent | Plasmid: pcMS2-NLS-Flag-CFI$_m$68ΔRS | This paper | 'CFIm68 Δ"' or '68ΔRS' | CFIm68 with the RS domain deleted inserted into pcMS2-NLS-Flag vector. See details in 'Plasmids' section of 'Materials and Methods' |
| Recombinant DNA reagent | Plasmid: pcMS2-NLS-Flag-CFI$_m$59 | This paper | 'CFIm59 F.L."', '59', 'MS2-NLS-fl-59' | CFIm59 inserted into pcMS2-NLS-Flag vector. See details in 'Plasmids' section of 'Materials and Methods' |

*Continued on next page*

*Continued*

| Reagent type (species) or resource | Designation | Source or reference | Identifiers | Additional information |
|---|---|---|---|---|
| Recombinant DNA reagent | Plasmid: pcMS2-NLS-Flag-CFI$_m$59ΔRS | This paper | 'CFIm59 Δ" or '59ΔRS' | CFIm59 with the RS domain deleted inserted into pcMS2-NLS-Flag vector. See details in 'Plasmids' section of 'Materials and Methods' |
| Recombinant DNA reagent | Plasmid: pcNMS2-NLS-Flag | *Sahin et al., 2010* | "-" | |
| Recombinant DNA reagent | Plasmid: pE-SUMO | LifeSensors | Cat#: 1001K | |
| Recombinant DNA reagent | Plasmid: pE-SUMO-CFIm25 | This paper | N/A | CFIm25 inserted into pE-SUMO vector. See details in 'Plasmids' section of 'Materials and Methods' |
| Recombinant DNA reagent | Plasmid: pcDNA3-Flag | *Sahin et al., 2010* | "-" | |
| Recombinant DNA reagent | Plasmid: Flag-CFI$_m$25 | This paper | "wt" | CFIm25 inserted into pcDNA3-flag vector. See details in 'Plasmids' section of 'Materials and Methods' |
| Recombinant DNA reagent | Plasmid: Flag-CFI$_m$25-m1 (Y158A/Y160A ) | This paper | "m1" | CFIm25 with Y158A/Y160A mutations inserted into pcDNA3-flag vector. See details in 'Plasmids' section of 'Materials and Methods' |
| Recombinant DNA reagent | Plasmid: Flag-CFI$_m$25-m2 (L218R) | This paper | "m2" | CFIm25 with L218R mutation inserted into pcDNA3-flag vector. See details in 'Plasmids' section of 'Materials and Methods' |
| Recombinant DNA reagent | Plasmid: Flag-CFI$_m$25-m3 (R63S) | This paper | "m3" | CFIm25 with R63S mutation inserted into pcDNA3-flag vector. See details in 'Plasmids' section of 'Materials and Methods' |
| Recombinant DNA reagent | Plasmid: pcDNA3 | Thermo Fisher | Cat#: V79020 | |
| Recombinant DNA reagent | Plasmid: pcΔ1,2 (B-A) | *Conrad et al., 2006* | N/A | |
| Recombinant DNA reagent | Plasmid: pcGFP-β1-MAT-E8-3′ | This paper | N/A | Used to create reporter cell line. See details in 'Plasmids' section of 'Materials and Methods' |
| Recombinant DNA reagent | Plasmid: hAAVS1-GFP-β2-MAT-E8-3′ | This paper | N/A | Used to create reporter cell line. See details in 'Plasmids' section of 'Materials and Methods' |
| Recombinant DNA reagent | Plasmid: hAAVS1-GFP-T2A-β2-MAT-E8-3′ hp2-6m9 | This paper | N/A | Use to create T2A, hp2-6m9 reporter cell lines. See details in 'Plasmids' section of 'Materials and Methods' |
| Recombinant DNA reagent | Plasmid: pSCRPSY | Clontech | Cat#V001595 | |
| Recombinant DNA reagent | Plasmid: pcEGFP | This paper (Materials and methods) | N/A | EGFP cloned into pcDNA. See details in 'Plasmids' section of 'Materials and Methods' |
| Recombinant DNA reagent | Plasmid: AAVS1 1L TALEN | Dr. Feng Zhang *Sajana et al., 2014* | RRID:Addgene#35431 | |
| Recombinant DNA reagent | Plasmid: AAVS1 1R TALEN | Dr. Feng Zhang *Sajana et al., 2014* | RRID:Addgene#35432 | |

*Continued on next page*

*Continued*

| Reagent type (species) or resource | Designation | Source or reference | Identifiers | Additional information |
|---|---|---|---|---|
| Recombinant DNA reagent | Plasmid: pEGFP-N1 | Clontech | Cat#6085–1 | |
| Recombinant DNA reagent | Plasmid: pAAVS-EGFP-DONOR | Dr. Joshua Mendell *Manjunath et al., 2019* | N/A | |
| Recombinant DNA reagree | Plasmid: LentiCRISPRv2 | Dr. Feng Zhang *Sajana et al., 2014* | RRID:Addgene#52961 | |
| Recombinant DNA reagent | Plasmid: pLentiV2-MAT2A | This paper | 'sgMAT2A' | sgMAT2A cloned into LentiCRISPRv2. See details in 'Plasmids' section of 'Materials and Methods' |
| Recombinant DNA reagent | Plasmid: pLentiV2-NT | This paper | 'sgNT' | sgNT cloned into LentiCRISPRv2. See details in 'Plasmids' section of 'Materials and Methods' |
| Recombinant DNA reagent | Plasmid: pX458-MAT2A-E9 | This paper | | Used to create 116-ΔDI cell line. See details in 'Plasmids' section of 'Materials and Methods' |
| Recombinant DNA reagent | Plasmid: pX459-MAT2A-E8 | This paper | | Used to create 116-ΔDI cell line. See details in 'Plasmids' section of 'Materials and Methods' |
| Recombinant DNA reagent | Plasmid: pSpCas9(BB)—2A-GFP (pX458) | Dr. Feng Zhang *Ran et al., 2013* | RRID:Addgene#48138 | |
| Recombinant DNA reagent | Plasmid: pSpCas9(BB)-Puro (pX459) | Dr. Feng Zhang *Ran et al., 2013* | RRID:Addgene#62988 | |
| Recombinant DNA reagent | Plasmid: pBS-ΔRI-Donor | This paper | N/A | Used to create 116-ΔDI cell line. See details in 'Plasmids' section of 'Materials and Methods' |
| Recombinant DNA reagent | Plasmid: pBluescript II SK + | Stratagene, La Jolla, California | Cat#212205 | |
| Recombinant DNA reagent | Plasmid library: Human CRISPR Knockout Pooled Library (Brunello) | Drs. David Root and John Doench *Doench et al., 2016* | RRID:Addgene#73179 | |
| Sequence-based reagent | Primers for Northern probes | This paper | N/A | See *Supplementary file 3* |
| Sequence-based reagent | Primers for RNase H cleavage | This paper | N/A | See *Supplementary file 3* |
| Sequence-based reagent | Primers for Making Plasmids | This paper | N/A | See *Supplementary file 3* |
| Sequence-based reagent | Insert for β-MAT-3′ UTR-Last-Motif-UGCA | GeneWiz | N/A | See *Supplementary file 3* |
| Sequence-based reagree | Insert for β-MAT-3 ′UTR-10/11 | GeneWiz | N/A | See *Supplementary file 3* |
| Sequence-based reagent | Negative Control No. 2 siRNA | Thermo Fisher | Cat#: 4390846 | *Silencer* Select |
| Sequence-based reagent | siMETTL16#1 | Thermo Fisher | Cat#: s35508 | *Silencer* Select |
| Sequence-based reagent | siMETTL16#2; siMETTL16 | Thermo Fisher | Cat#: s35507 | *Silencer* Select |
| Sequence-based reagent | siCFIm25#1 | Thermo Fisher | Cat#: s21770 | *Silencer* Select |
| Sequence-based reagent | siCFIm25#2 | Thermo Fisher | Cat#: s21772 | *Silencer* Select |
| Sequence-based reagent | siCFIm68#1 | Thermo Fisher | Cat#: s21773 | *Silencer* Select |

*Continued on next page*

*Continued*

| Reagent type (species) or resource | Designation | Source or reference | Identifiers | Additional information |
|---|---|---|---|---|
| Sequence-based reagent | siCFIm68#2 | Thermo Fisher | Cat#: s21774 | *Silencer* Select |
| Sequence-based reagent | siCFIm59#1 | Thermo Fisher | Cat#: s224836 | *Silencer* Select |
| Sequence-based reagent | siCFIm59#2 | Thermo Fisher | Cat#: s224837 | *Silencer* Select |
| Sequence-based reagent | siCFIm73#1 | Thermo Fisher | Cat#: s28531 | *Silencer* Select |
| Sequence-based reagent | siCFIm73#2 | Thermo Fisher | Cat#: s28532 | *Silencer* Select |
| Peptide, recombinant protein | rSUMO-CFI$_m$25 | This paper | N/A | |
| Commercial assay or kit | CellTiter-Glo | Promega | Cat#G7570 | |
| Commercial assay or kit | AMPure XP | Beckman Coulter | Cat#A63880 | |
| Chemical compound, drug | MG132 | Sigma | Cat # M8699-1MG | 50 µM |
| Software, algorithm | ImageQuant (v 8.1) | GE Healthcare Life Sciences | N/A | |
| Software, algorithm | Graphpad Prism (v 8) | GraphPad Software | RRID:SCR_000306 | |
| Software, algorithm | GelQuant.NET (v 1.8.2) | *BiochemLab Solutions, 2011* | N/A | |
| Software, algorithm | RStudio (v 3.5.1) | *RStudio Team, 2018* | N/A | |
| Software, algorithm | Image Studio (v 5.2) | LI-COR | RRID:SCR_015795 | |
| Software, algorithm | SuperExactTest | *Wang et al., 2015* | doi:10.1038/srep16923 | |
| Software, algorithm | MAGeCK | *Li et al., 2014* | https://sourceforge.net/projects/mageck/files/ | |
| Software, algorithm | Fastqc | *Andrews, 2019* | https://www.bioinformatics.babraham.ac.uk/projects/fastqc/ | |
| Software, algorithm | Fastp (v 0.19.5) | *Chen et al., 2018* | doi:10.1093/bioinformatics/bty560 | |
| Software, algorithm | Cutadapt (v 1.18) | *Martin et al., 2012* | https://cutadapt.readthedocs.io/en/stable/ | |
| Software, algorithm | HISAT2 (v 2.1.0) | *Kim et al., 2015* | https://www.ncbi.nlm.nih.gov/pmc/articles/PMC4655817/ | |
| Software, algorithm | DESeq2 (v 1.22.1) | *Love et al., 2014* | doi:10.1186/s13059-014-0550-8 | |
| Software, algorithm | DPAC (v 1.10) | *Routh, 2019a Routh, 2019b* | https://sourceforge.net/projects/dpac-seq/ | |
| Software, algorithm | FloJo (v 9.9.5) | FlowJo LLC | N/A | |
| Software, algorithm | CellCapTure (v3.1) | *Stratedigm, Inc., 2021* | N/A | |

## Resource availability

### Lead contact

Further information and reagent requests may be directed to Nicholas K. Conrad (Nicholas.Conrad@UTSouthwestern.edu).

### Materials availability

Plasmids generated in this study are available upon request of the lead contact, Nicholas K. Conrad (Nicholas.Conrad@UTSouthwestern.edu).

## Experimental models and subject details

### Cell culture

HEK293 and HEK293T cells were maintained in DMEM (Sigma, Cat#D5796) with penicillin-streptomycin, 2 mM L-glutamine, and 10% fetal bovine serum (FBS, Sigma, Cat#F0926) and grown at 37°C in 5% $CO_2$. The same media was used with a maintenance concentration of Plasmocin (InvivoGen, Cat#ant-mpt, 1:10,000) for all HCT116-based cell lines and 50 µg/ml hygromycin for reporter cell lines. 293A-TOA cells were cultured similarly to HEK293 cells, but with Tet-free FBS (Atlanta Biologicals, Cat#S10350) and 100 µg/ml G418. Care was taken to ensure cells were passaged in methionine-rich media to keep MAT2A DI/mRNA ratios consistent between experiments. Methionine-free media DMEM (Thermo Fisher, Cat#21013024) was supplemented with 0.4 mM L-cysteine and 1 mM sodium pyruvate in addition to penicillin-streptomycin, L-glutamine, and Tet-free FBS. All cell lines have been validated by STR analysis and are routinely tested for mycoplasma.

### Generation of GFP-reporter lines

HCT116 cells were co-transfected in a 6-well plate with 0.2 µg AAVS1 1L TALEN, 0.2 µg AAVS1 1R TALEN, and 1.6 µg hAAVS1-GFP-β2-MAT-E8-3′. The next day cells were split to 10 cm plates, and hygromycin was added to 100 µg/ml. Cells were selected for a total of two weeks, initially in 100 µg/ml hygromycin and then in 250 µg/ml hygromycin over the second week. Fluorescence-activated cell sorting (FACS) was used to select clonal cell lines with low to mid GFP output in methionine-rich conditions. Clonal cell lines were selected based on high differential GFP expression between methionine-rich and methionine-starved conditions. The cell line that provided the most robust GFP expression after methionine depletion was chosen as the reporter line. As described (*Figure 1—figure supplement 1*), the robust differential results in part from an alternate splicing pattern that stabilizes the GFP protein.

The modified GFP reporter cell lines containing a T2A element and mutations in hp2-6 to abrogate METTL16 activity were created in a similar fashion to the original reporter, with the only change being using hAAVS1-GFP-T2A-β2-MAT-E8-3 ˙hp2-6m9 instead of hAAVS1-GFP-β2-MAT-E8-3′.

### Generation of 116-ΔDI line

Two 10 cm plates of HCT116 cells were each transfected with 3 µg of pX458-MAT2A-E9, 3 µg of pX459-MAT2A-E8, and 10 µg of pBS-ΔRI-Donor. Eight hrs later, fresh media was added that included 1 µg/ml puromycin and 1 µM of the NHEJ inhibitor SCR7 (Fisher Scientific; *Chu et al., 2015*; *Maruyama et al., 2015*). Approximately 48 hr after transfection, puromycin-resistant transfected cells were subjected to FACS, and GFP-positive single cells were seeded on a 96-well plate. SCR7 was included for an additional 3–5 days. After clonal expansion, DNA was harvested, and MAT2A DI status was examined by PCR using primers NC1145 and NC2537. The sequences of all primers and DNA oligonucleotides are listed in *Supplementary file 3*. Only one clonal line contained DI deletions on both alleles. Subsequent sequencing demonstrated that one allele had the DI deleted by HR while the other allele had intron eight deleted by NHEJ. The latter deleted an additional 20 nt (18 nt from exon 8 and 2 nt from exon 9) to create an out-of-frame junction between exons 8 and 9: CGA TCT CCG/**AT CTG GAT** (exon8/**exon9;** see also *Figure 3A*).

## Method details

### Methionine depletion

Cells were transferred into fresh DMEM media supplemented with an additional 200 µM methionine the day before depletion. To deplete methionine, cells were washed twice with phosphate buffered saline (PBS) supplemented with calcium chloride and magnesium chloride (Sigma) before replacing with growth media containing or lacking methionine as required.

### Transfection

Cells were transfected using TransIT-293 (Mirus, Cat#MIR2706) according to the manufacturer's protocol. For co-immunoprecipitations, 600 µl Opti-MEM (Thermo Fisher, Cat#31985–070) and 36 µl TransIT-293 were incubated at room temperature (RT) for 5 min before addition to 10 µg of aliquoted DNA. The mixture was incubated 15 min before adding dropwise to a 10 cm dish of cells. For typical 12-well transfections, 2 µl TransIT-293, 40 µl Opti-MEM, and 800 ng plasmid was added per well using the above protocol. MS2-tethering experiments used 30 ng of reporter, 200 ng of MS2-NLS-flag tagged construct, and 570 ng of pcDNA3 per well of a 12-well plate. Expression of MS2-NLS-flag protein was analyzed after transfecting 400 ng of MS2-NLS-flag construct and 400 ng pcDNA per well. For experiments in *Figures 4*, 100 ng of reporter construct and 700 ng pcDNA3 was transfected per well of a 12-well plate into HEK293 cells. For *Figure 4—figure supplement 1*, 15 ng of reporter and 785 ng pcDNA3 was transfected on day three of a 4-day knockdown in 293A-TOA cells. In some cases, HCT116 cells were transfected using FuGENE HD Transfection Reagent (Promega, Cat#E2311) according to the manufacturer's protocol.

## RNA extraction and purification

RNA was harvested using TRI Reagent (Molecular Research Center, Inc, Cat#TR118) with minor variations to manufacturer's protocol. For one well of a 12-well plate, 400 µl of TRI Reagent was added. Upon homogenization by pipetting, 80 µl chloroform was added to the TRI Reagent containing tube before shaking vigorously by hand until homogenization. The samples were centrifuged at 12,000 x g for 15 min at 4°C before transferring the aqueous phase to a fresh tube then mixing with an equal volume of chloroform. Care was taken not to disturb the interphase between the organic and aqueous phases. The chloroform/aqueous mixture was shaken vigorously by hand and centrifuged at 12,000 x g for 5 min at RT before transferring the aqueous phase once again to a fresh tube. The aqueous solution was mixed with an equal volume of isopropanol for storage at –20°C or precipitation by centrifugation at 16,000 x g 10 min RT with the addition of 15 µg GlycoBlue Coprecipitant (Thermo Fisher, Cat#AM9516).

If necessary, RNA was further purified by an additional phenol:chloroform:iso-amyl alcohol (PCA, 25:24:1) extraction step, in which an equal volume of PCA was added to RNA in an aqueous solution. The mixture was vortexed briefly before centrifugation at 14,000 x g 5 min RT. The aqueous phase was transferred to a fresh tube and mixed with an equal volume of chloroform. The samples were mixed by vigorous shaking before a second centrifugation step at 14,000 x g 5 min RT. The aqueous phase was mixed 15 µg glycoblue and 2.5X volumes of ethanol before precipitation at –80°C for 30 min or storage at –20°C.

## siRNA knockdown

293A-TOA or HCT116 lines and their derivatives were transfected with 30 nM siRNA using RNAi-MAX following the manufacturer's protocol. Twenty-four hr after transfection, confluent cells were split to allow for an additional 3 days of cell division and knockdown (total 96 hr knockdown). Degree of cell dilution when passaging at this stage was dependent on the toxicity associated with the specific knockdown. We changed the media 24 hr before harvesting to ensure cells were maintained in a methionine-rich conditions. For samples using multiple siRNAs targeting the same gene, 15 nM of each siRNA was used for a total of 30 nM. For knockdown prior to transfection (*Figure 7* and *Figure 4—figure supplement 1*), cells were transfected with the appropriate tethering or reporter constructs 72 hr post knockdown after a media change to ensure cells were harvested in methionine-rich conditions. Cells were harvested 24 hr post-transfection and 96 hr post-knockdown.

## Poly(A) selection

Sera-Mag Oligo(dT)-Coated Magnetic particles (GE Healthcare Lifesciences, Cat#38152103010150) were washed three times in 1XSSC (150 mM NaCl, 15 mM sodium citrate) with 0.1% SDS then resuspended in the same using the initial volume. Purified total RNA in water was heated at 65°C for 5 min before the addition of SSC and SDS to 1X and 0.1% respectively. Washed Sera-Mag Oligo(dT)-Coated Magnetic particles were added to the RNA (20 µl particles per 40 µg total RNA). Samples were nutated 20 min at RT then washed three times in 0.5X SSC/0.1% SDS. RNA was eluted in 100 µl water for 5 min at RT, with gentle aggitation. The supernatant was combined with a second elution in 100 µl water for 5 min at 65°C. The combined eluants were further purified by PCA and chloroform extraction followed by ethanol precipitation as described above.

## RNase H mapping

Poly(A) selected RNA purified from an initial 160 µg total RNA in $H_2O$ was divided equally into three tubes. Five µM of specific DNA oligonucleotide and 1 µM $dT_{20}$ was added, then samples were diluted to 10 µL reaction volumes before incubation at 65°C for 5 min. After cooling on ice 3 min, the following was added to each tube to reach a final volume of 20 µl per reaction: RNase H buffer (20 mM Tris pH7.5, 100 mM KCl, 10 mM $MgCl_2$ final concentration), 10 mM DTT, 0.75 U RNase H, and 16U RNasin Plus. Samples were digested for 1 hr at 37°C before quenching with 180 µl G50 buffer (20 mM Tris pH7, 300 mM sodium acetate, 2 mM EDTA, 0.25% SDS), then purified using PCA and chloroform extractions followed by ethanol precipitation as described above.

## Northern blotting

Northern blots were performed using standard techniques and probed with radiolabeled RNA transcripts (*Ruiz et al., 2019*). RNA probes were produced using a digested plasmid or PCR products containing a T7 or SP6 RNA polymerase promoter. Primer and plasmids are listed in *Supplementary file 3*. Either 3–5 µg total RNA or polyadenylated enriched RNA produced from 20 to 40 µg total RNA was loaded per lane.

## Coimmunoprecipitation

Plasmids expressing flag-tagged proteins were transfected into 10 cm dishes of HEK293 cells. After washing twice, cells were scraped in PBS. Cells were pelleted at 1000 x g 4°C for 3 min. PBS was removed and cells were lysed in RSB100T (50 mM Tris pH 7.5, 100 mM NaCl, 2.5 mM $MgCl_2$, 1 mM $CaCl_2$, 1% TritonX100) with 1 mM PMSF and 1X Protease Inhibitor Cocktail Set V (Millipore, Cat#539137). Samples were nutated at RT with 20U RQ1 RNase-free DNase (Promega, Cat#M6101) and RNase A (10 µg/ml) for 15 min before clarifying twice by centrifugation at 4°C 21,000 x g for 10 min. Lysate was bound to pre-washed ANTI-FLAG M2 Affinity Gel (Sigma, Cat#A2220) by nutating at 4°C for 2 hr before washing five times with RSB100T. Protein was eluted by vortexing for 30 min at 4°C in RSB100T supplemented with 0.4 mg/ml 3X FLAG peptide (Sigma, Cat#F4799), 1 mM PMSF, and 1X Protease Inhibitor Cocktail Set V. Samples were analyzed by standard western blotting protocols.

## SAM metabolite extraction

Protocol was adapted from *Dettmer et al., 2011* and *Tu et al., 2007*. Cells were treated with siRNAs as indicated and maintained in methionine-rich conditions in 10 cm plates. Ninety-six hr after knockdown, cells were washed three times with ice-cold PBS with calcium chloride and magnesium chloride before the addition of 1200 µl ice-cold 80% methanol while being snap frozen in liquid nitrogen. Samples were scraped on ice, transferred to Eppendorf tube after mixing by pipetting, then frozen in liquid nitrogen. Samples were thawed in a RT water bath while mixing by pipetting before clarification at 16,000 x g 4°C for 10 min. Methanol supernatants were stored at −80°C and cell pellets were washed 1X PBS, resuspended in 1X SDS PAGE loading buffer without loading dye (2% SDS, 62.5 mM Tris pH 6.8, 10% glycerol, 1% β-mercaptoethanol) then sonicated until homogenous. Relative protein concentration measured by nanodrop was used to estimated cell number between samples and methanol supernatant volumes were adjusted accordingly. Methanol supernatants were dried using a speed vacuum before resuspension in Solvent A (1% formic acid in water), centrifuged twice, then filtered through 0.2 µM PVDF to remove insoluble particles. Samples were

analyzed via LC-MS/MS with a total run time of 20 min, flow rate of 0.5 ml/min, 0.1% formic acid in water as Solvent A, and 0.1% formic acid in methanol as Solvent B. Pure SAM was injected and analyzed alongside samples for each experiment. SAM was detected by multiple reaction monitoring (MRM) using the ion pair 339/250, quantified using the Analyst 1.6.1 Software package by calculating total peak area, then normalized to non-targeting control (*Dettmer et al., 2011*; *Tu et al., 2007*).

## Flow cytometry

HCT116 reporter cells were conditioned in methionine-rich or methionine-free media for 24 hr before harvesting by trypsinization. After quenching the trypsin, cells were pelleted by centrifugation at 800 x g for 3 min at 4°C then washed with PBS. After washing, cells were resuspended in 5% formaldehyde in PBS and nutated 1 hr to overnight. Prior to analysis by flow cytometry, cells were pelleted at 800 x g for 3 min at 4°C then resuspended in PBS with 3% FBS. Samples were aliquoted into 96-well v-bottom dishes to be analyzed on a Stratedigm S1000. Flow cytometry data was analyzed by FloJo to compare relative GFP fluorescence.

## CRISPR screen

The Human Brunello CRISPR knockout pooled library, a gift from David Root and John Doench (Addgene #73179), was prepared according to the BROAD institute pDNA library amplification protocol (*BROAD Institute Amplification of pDNA libraries Protocol, 2021*; *Doench et al., 2016*; *Shalem et al., 2014*). In brief, 400 ng of library was electroporated into ElectroMAX Stbl4 Competent Cells (Thermo Fisher, Cat#11635018). After recovery in SOC Outgrowth Medium (New England Biolabs, Cat#B9020S), the sample was plated equally between four 500 cm$^2$ bioassay plates containing LB agar with 100 µg/ml puromycin using a biospreader (Bacti Cell Spreader, VWR International). Cells were grown 18 hr at 37°C before scraping with ice cold LB. DNA was prepared by dividing the total cell mass evenly between two Qiagen maxi prep columns, following the manufacturer's protocol.

To produce lentivirus, twenty 15 cm tissue culture plates were coated with poly-D-lysine (100 ug/ml in milliQ water, 0.22 µM filtered) for 5 min then washed twice with PBS before plating 293 T cells. pMD2.G and psPAX2 were a gift from Didier Trono (Addgene plasmid #12259 and #12260). Brunello library, pMD2.G, and psPAX2 were co-transfected between the twenty plates equally, using a total of 300 µg, 120 µg, and 180 µg plasmid, respectively. Prior to transfection, the media was exchanged for DMEM supplemented with 3% FBS instead of the normal 10%, with subsequent media used for the production of lentivirus likewise containing 3% FBS. Media was changed 6 hr post transfection, then collected and pooled at 48 and 72 hr post-transfection. HEPES pH 7.2 (20 mM) and 4 µg/ml polybrene was added to the lentivirus before filtration through a 0.45 µM filter to clear debris. The filtered virus was treated with Benzonase Nuclease (Sigma, Cat#E1014) to digest residual plasmid (50 U/ml benzonase, 50 mM Tris pH 8.0, 1 mM MgCl$_2$, 0.1 mg/ml BSA) for 30 min at 37°C with gentle agitation. Aliquoted lentivirus was snap frozen in liquid nitrogen then stored at −80°C until further use.

Lentivirus titer was determined by transduction of HCT116 reporter cells plated at 4 × 10$^5$ cells per well of a 6-well plate. Cells were split evenly into media ± 1 µg/ml puromycin 48 hr post-transduction. Five days after transduction, cells were harvested and analyzed via CellTiter-Glo Luminescent Cell Viability Assay (Promega, Cat#G7570). To estimate the viral titer, we compared cell counts in selected vs unselected conditions for each lentivirus treatment condition.

For the genome-wide screen, a titer that infected ~20% of plated cells was used. To obtain 100X coverage of the 76,441 gRNA in the Brunello library, sixteen 6-well plates with 4 × 10$^5$ cells per well were infected. Two days post transduction, cells were split into media containing 1 µg/ml puromycin and Plasmocin (1:10,000). Media was changed every day and cells were split as needed, with 200 µM additional methionine added on day 7 post-transduction. On day eight, cells were deprived of methionine 18 hr before sorting by FACS. Care was taken to sort cells within an 18–20 hr window after methionine depletion to maintain consistency between replicates.

To prepare cells for FACS, cells were trypsinized then diluted in ice-cold PBS, pipetting gently yet thoroughly to ensure cell clumps were broken up. Cells were pelleted at 1600 x g for 3 min at 4°C before resuspension in FACS buffer (50 mM HEPES pH 7.2, 0.5 mM EDTA, 2% Tet-free FBS, PBS).

Resuspended cells were strained using a 100 μM nylon cell strainer (Falcon 100 μM Cell Strainer, Cat#352360) and kept on ice before and during sorting. Cells with the lowest 1% of GFP signal were collected in FACS collection buffer (50% Tet-free FBS, 50 mM HEPES pH 7.2, PBS) with two rounds of enrichment using a BD Biosciences FACSAria Fusion Cell Sorter. The first round of enrichment of cells sorted at approximately 12,000 cells/sec. However, due to the droplet size, several mid to high GFP expressing cells appeared in the sample. The second round of enrichment removed the mid to high GFP expressing cells, only accepting cells within the initial gating set for the lowest 1% of GFP-expressing cells relative to the unsorted population. For unsorted input to compared to the GFP-depleted sample, a minimum of 8 million cells were set aside for lysis without sorting. Sample coverage was calculated by the number of cells collected in the final sample divided by 1% of the number of gRNA found in the Brunello library (*Equation 1*). Cells were centrifuged at 1600 x g for 3 min 4°C, resuspended in 500 μl tissue lysis buffer (10 mM Tris pH 8, 100 mM NaCl, 11 mM EDTA, 200 μg/ml Proteinase K, 0.4% SDS) and lysed at 55°C 550 rpm overnight.

*Equation 1*. Brunello library coverage.

$$\frac{Cells\ collected\ in\ sorted\ sample}{\left(0.01\ \frac{lowest\ GFP}{population}\right) \times \left(76,441\ \frac{gRNA}{library}\right)} \tag{1}$$

To isolate DNA, the lysates were cooled 3 min at RT before the addition of 5 μl 2 mg/ml RNase A. After briefly vortexing, samples were shaken at 37°C 550 rpm for 1 hr. An equal volume of PCA was added per tube before vortexing 20 s and dividing equally between two phase lock tubes precentrifuged at 16,000 x g for 30 s (1.5 ml MaXtract High Density Tubes, Qiagen, Cat#129046). Samples were centrifuged at 16,000 x g for 5 min RT before an additional 500 μl PCA was added to each phase lock tube, without transferring the aqueous layer. Tubes were inverting vigorously for ~1 min to mix then centrifuged at 16,000 x g for 5 min RT. The aqueous phase was then transferred to a fresh, precentrifuged phase lock tube with the addition of 500 μl chloroform. The tubes were once again inverted vigorously before centrifuging at 16,000 x g for 5 min RT. The aqueous phase was transferred to a fresh Eppendorf tube with the addition of 20 μg GlycoBlue Coprecipitant. DNA was precipitated in ethanol at −80°C, pelleted at 16,000 x g RT for 10 min before being washed with 75% ethanol. DNA pellets were resuspended in sterile water at RT for 30 min. After adequate time for resuspension, the DNA was pipetted ~20 times to shear the DNA.

Library amplification was completed following a two-step variation of the BROAD institute protocol (*BROAD Institute PCR of sgRNAs for Illumina sequencing Protocol, 2021*). The two-step variation of the BROAD institute protocol consists of an initial amplification step using primers flanking the P5 and P7 primers used for sequencing. The initial PCR product (PCR#1) is then diluted to be used as template for amplification Illumina P5/P7 flow cell primers (PCR#2). For the first PCR step, 6.6 μg DNA was used as template per 100 μl reaction, with twenty reactions being set up per sample if possible. If the sample contained less than 13.2 μg DNA, the DNA was divided evenly between a minimum of two reactions. The DNA was amplified using with TaKaRa ExTaq and primers NC3196 and NC3197 flanking the P5 and P7 sites (1X ExTaq reaction buffer, 0.2 mM each dNTP, 0.5 μmol each primer, 0.075U ExTaq polymerase, 6.6 μg template DNA, water to 100 μl per reaction). PCR#1 consisted of 18 cycles following the ExTaq manufacturer's protocol (95°C 1 min, [95°C 30 s, 53°C 30 s, 72°C 30 s] x 18, 72°C 10 min, hold 4°C). For the second PCR amplification step, 5 μl of the pooled PCR#1 reaction was used as template for each 100 μl PCR#2 reaction. Four 100 μl reactions were made per sample using the reaction conditions above, except replacing the primers NC3196 and NC3197 with Illumina P5 stagger primer mix and a P7 barcode primer (*Supplementary file 3*). The P5 stagger primer consisted of all eight stagger primers evenly mixed. A unique P7 barcode primer was used for each sample. The PCR cycle number for PCR#2 was selected to match band intensity between samples when run on an agarose gel (8–12 cycles).

The 400 μl PCR#2 product per sample was pooled before purification by AMPure XP (Beckman Coulter, Cat#A63880) following the BROAD institute protocol (BROAD Institute PCR of sgRNAs for Illumina sequencing). Briefly, the pooled PCR volume was mixed with 0.5 AMPure XP bead volume then incubated 5 min RT. Beads were separated on a magnetic strip for 2 min then the supernatant transferred to a fresh tube to be mixed with 1.8 volumes of AMPure XP beads. After a 5-min RT incubation, beads were separated for 3 min on a magnetic strip before washing twice with 70% EtOH for 1 min each while remaining on the magnet. The beads were dried for 5 min before eluting with

the starting volume of water (400 µl) for 2 min at RT. The beads were separated from the sample for 2 min on the magnetic strip, and the eluted DNA was transferred to a fresh tube. The purification protocol was repeated a second time, with the only change being a final elution volume of 50 µl water instead of 400 µl.

Amplified library was analyzed by Qbit, TapeStation, and qPCR to assess library purity and concentration before sequencing. Three independent biological replicates were sequenced on an Illumina NextSeq 500 with read configuration as 75 bp, single end. The fastq files were subjected to quality check using fastqc (version 0.11.2, http://www.bioinformatics.babraham.ac.uk/projects/fastqc) and fastq_screen (version 0.4.4, http://www.bioinformatics.babraham.ac.uk/projects/fastq_screen), and adapters trimmed using an in-house script. The Human Brunello CRISPR library sgRNA sequences were downloaded from Addgene (https://www.addgene.org/pooled-library/). The trimmed fastq files were mapped to Brunello library sequence with mismatch option as 0 using MAGeCK (*Li et al., 2014*). Further, read counts for each sgRNA were generated and median normalization was performed to adjust for library sizes. Positively and negatively selected sgRNA and genes were identified using the default parameters of MAGeCK.

## Poly(A)-ClickSeq

RNA was extracted from HCT116 or 116-ΔDI cells grown in methionine-rich conditions and treated with non-targeting or CFI$_m$25 siRNA. RNA was treated with RQ1 RNase-free DNase with the addition of RNasin Plus RNase Inhibitor (Promega, Cat#N2615)(DNAse RQ1 1X reaction: pH 8.0, 10 mM NaCl, 6 mM MgCl$_2$, 10 mM CaCl$_2$, 5U RQ1 DNase, 40U RNasinPlus) for 30 min at 37°C before PCA and chloroform extraction followed by ethanol precipitation. Precipitated RNA was analyzed by TapeStation to verify that the RNA samples had a RIN of 8.5 or greater before submission.

Poly(A)-ClickSeq was performed by ClickSeq Technologies, Galveston, TX, using methods described elsewhere (*Elrod et al., 2019*; *Routh, 2019b*; *Routh et al., 2017*). Briefly, samples were prepared by RT-PCR using an oligo-dT with a P7 adapter. The P5 primer was attached using click-chemistry before amplifying the RT product. The samples were sequenced on an Illumina NextSeq 550 using a Mid Output 130M, v2 flow cell. Only reads with >40 nts of cDNA sequence and >10 A's were retained for analysis to ensure mapping of the 3′UTR. The reads were quality filtered then aligned, with most localization to the 3′UTR of transcripts. Sites with >5 reads within 10nts were defined as a poly(A) cluster (*Elrod et al., 2019*; *Routh et al., 2017*). Samples were analyzed using Differential Poly(A)-Clustering (DPAC)(*Routh, 2019b*).

## Purification of recombinant SUMO-CFI$_m$25

Rosetta (DE3) cells (EMD Millipore) were transformed with pE-SUMO-CFI$_m$25 and selected in 30 µg/ml chloramphenicol and 50 µg/ml kanamycin. Colonies were inoculated into a 2 mL starter culture and grown at 37°C overnight. The culture was diluted into 200 mL fresh LB media with antibiotics, grown to mid-log phase (O.D. ~0.5), and IPTG was added to 1 mM. After 2 hr at 37°C, bacterial pellets were harvested by 10 min centrifugation at 4000 x g and the pellets were resuspended in 1 mL lysis buffer (300 mM NaCl, 50 mM NaH$_2$PO$_4$, 0.5% Triton X-100, 0.5 mM PMSF, pH 8.0). Two milligrams lysozyme (Sigma) was added and the mix was incubated on ice for 30 min. Benzonase was then added to 0.25 U/mL and the mix was nutated for 30 min at RT before the lysate was cleared by centrifugation at 10,000 x g for 30 min at 4°C. The protein was purified in batch by incubation with 1 mL of Ni-NTA Agarose (QIAGEN, Cat#30210) for 1 hr at 4°C. The beads were washed over a column with 10 volumes of wash buffer (300 mM NaCl, 50 mM NaH$_2$PO$_4$, 10 mM imidazole, 0.5% Triton X-100, 0.5 mM PMSF, pH 8.0) and proteins were collected in fractions of elution buffer (300 mM NaCl, 50 mM NaH$_2$PO$_4$, 250 mM imidazole, pH 8.0). Eluted fractions were pooled and dialyzed twice at 4°C in 2L of Buffer D (20 mM Hepes pH 7.9, 20% glycerol, 50 mM KCl, 0.2 mM EDTA, 0.5 mM DTT) using a Slide-A-Lyzer Dialysis Cassette (10,000 kD cutoff; Fisher). The first dialysis step was for 2 hr, buffer was replaced, and the second dialysis step occurred overnight. The samples were concentrated ~2-fold using Amicon Ultra 0.5 centrifugal filter units to a final concentration 0.8 mg/mL. Aliquots of the protein were stored at −80°C.

## Label transfer assays

The RNAs used as substrates and competitors were synthesized by Sigma (*Supplementary file 3*). The substrates were 5′-end labeled using T4 polynucleotide kinase and gamma-$^{32}$P-ATP. For the label transfer reaction without competitors, recombinant SUMO-CFI$_m$25 was first treated with protease Ulp1 (LifeSensors, Cat#SP4010) for one hour at 30°C (~7 units of protease per 100 µg of SUMO-CFI$_m$25). Cleaved or untreated SUMO-CFI$_m$25 was then used in 10 µL binding reactions at a final concentration of 3 µM with 2 nM of the radiolabeled RNA substrates as well as 0.75% polyvinyl alcohol (PVA), 15 mM Tris-HCl pH 8, 15 mM NaCl, 25 mM KCl, 5 mM DTT, 11% glycerol, 10 mM HEPES, 0.1 mM EDTA, 0.25 µg *E. coli* tRNA, and 10 units RNasin. The reactions were incubated at 30°C for 15 min. For the competition assays, 10 µL binding reactions were performed with 1 µM SUMO-CFI$_m$25 and 2 nM radiolabeled WT/UGUU RNA with the indicated concentrations of cold RNA competitor that had been treated at 70°C for 5 min then placed on ice. The reaction also included 0.75% PVA, 15 mM Tris-HCl pH 8, 40 mM KCl, 5 mM DTT, 7.5% glycerol, 1 mM HEPES, 0.01 mM EDTA, 0.25 µg *E. coli* tRNA, and 10 units RNasin. The reaction was incubated at 30°C for 30 min. All samples were crosslinked at 860 mJ/cm$^2$ on ice ~2 cm from a 254 nm UV light source (Spectroline XL-1500). The samples were resolved by SDS-page, the gel was dried and then bands analyzed by Phosphorimager.

## Plasmids

Plasmids pX458-MAT2A-E9 and pX459-MAT2A-E8 are derived from pSpCas9(BB)−2A-GFP (pX458) and pSpCas9(BB)−2A-Puro (pX459) V2.0 which were gifts from Dr. Feng Zhang (Addgene plasmids #48138 and #62988). The targeting sequence was inserted using BbsI digestion and annealed 5′ phosphorylated oligonucleotides NC2980 and NC2981 (*Supplementary file 3*; pX458-MAT2A-E9) or NC2978 and NC2979 (*Supplementary file 3*; pX459-MAT2A-E8) as previously described (*Ran et al., 2013*). To make pBS-ΔRI-Donor, we amplified the left homology arm with primers NC2982 and NC2983 (*Supplementary file 3*) and the right homology arm with primers NC2984 and NC2985 (*Supplementary file 3*) using cDNA from methionine-starved cells as a template to ensure PCR products had no intron 8. We used SOEing to join these products (*Horton, 1995*) and then inserted them into pBluescript II SK +using NotI and BamHI restriction sites.

MAT2A β-globin reporter variants were made using β-MAT-WT from *Pendleton et al., 2017*. For the UGUU to UG**C**U mutation between the stop codon and hp1, β-MAT-WT was digested with XbaI before two PCR fragments amplified by NC3289 and NC3290 or NC3287 and NC3288 were inserted via Gibson Assembly Cloning kit (New England Biolabs)(*Supplementary file 3*). The UGUU to UGU**A** mutation was likewise made by Gibson assembly. First, NC2935 and NC1747 or NC3700 and NC3346 were used to amplify β-MAT-WT. Then the two PCR products were inserted via Gibson assembly into β-MAT digested with EcoRI and XhoI. The DI mutations (UG**C**A and **C**GUA) were made in a manner similarly to the UGUU to UGU**A** mutation. Two PCR products (see *Supplementary file 3* for primers) were amplified from β-MAT-WT then inserted into β-MAT-WT digested with EcoRI and XhoI via Gibson assembly. Mutations to the UGUA motifs found in the MAT2A 3′UTR were produced by amplifying a synthesized DNA fragment (GeneWiz) with NC3667 and NC3668 then insertion by Gibson assembly into β-MAT-WT digested with ApaI.

A plasmid containing the CFI$_m$25 cDNA was obtained from the McDermott Sanger Sequencing Core. CFI$_m$25 was amplified by NC3604 and NC3752 and then inserted into the pcDNA-flag and pcMS2-NLS-flag vectors using BamHI and XhoI sites. Similarly, CFI$_m$25 derivatives were produced using SOEing using NC3604 and NC3752 as the exterior flanking primers and inserted into the same vectors using BamHI and XhoI sites. The internal primers used for mutagenesis are listed in *Supplementary file 3*.

CFI$_m$68 and CFI$_m$59 were amplified as two fragments each from cDNA produced by oligo-dT priming, using primers listed in *Supplementary file 3*. The fragments were joined by SOEing using the forward primer from PCR#1 and reverse primer from PCR#2 then ligated into pcMS2-NLS-flag using BamHI and XhoI sites. Constructs with the RS domain deleted were produced by amplification from the pcMS2-NLS-fl-CFI$_m$68 or -CFI$_m$59 vectors using primers listed in *Supplementary file 3* before insertion into pcMS2-NLS-flag using BamHI and XhoI sites. pLentiV2-MAT2A was produced by annealing NC2533 and NC2534 before insertion into LentiCRISPRv2 (Addgene #52961; gift of Dr. Feng Zhang) following the Zhang lab protocol (*Ran et al., 2013*; *Sanjana et al., 2014*;

*Shalem et al., 2014*). pLentiV2-NT was produced similarly, by annealing NC3198 and NC3199 before insertion into LentiCRISPRv2.

The pE-SUMO-CFI$_m$25 expression plasmid was generated by amplification of CFI$_m$25 cDNA with NC3852 and NC3853. The PCR product was digested with BbsI and XbaI and ligated into pE-SUMO cut with BsaI.

The construct pcbD1-MAT-E8-3′ HP1AG, (2xMS2; β-MAT-hp1G4, 2XMS2) was generated by SOE-ing joining amplicons generated with primers NC1576 and NC2671 with those from SP6 +and NC2670. PCR reactions used pcβD1-MAT-E8-3′ HP1AG (No MS2) plasmid as a template. The products were inserted into β-MAT-WT cut with EcoRI and XhoI.

AAVS1 1L TALEN and AAVS1 1R TALEN were gifts from Dr. Feng Zhang (Addgene plasmids #35431 and #35432)(*Sajana et al., 2014*). The plasmid hAAVS1-GFP-β2-MAT-E8-3′ was generated in two steps. First, we made pcGFP-β1-MAT-E8-3′ by Gibson assembly of three DNA fragments using the Gibson assembly. One insert was generated by amplifying EGFP using primer pair NC2229/NC2230 with pEGFP-N1 as a template and the second insert was made with primer pair NC2231/NC2232 using β-MAT-WT as a template. The vector fragment was generated by gel purification of β-MAT-WT cut with HindIII. The resulting plasmid was used as a template for PCR amplification with primers NC2264/NC2265 and the product was inserted into pAAVS-EGFP-DONOR digested with FseI and XbaI by Gibson assembly to generate hAAVS1-GFP-β2-MAT-E8-3′. The plasmid hAAVS1-GFP-T2A-β2-MAT-E8-3 ′hp2-6m9 was made in two steps. First, we made pcGFP-T2A-β2-MAT-E8-3 ′mhp2-6 by Gibson assembly of three DNA fragments. The vector fragment was pcEGFP, which was made by amplification of eGFP from pEGFP-N1 using primers NC3272 and NC3273 and insertion of the product into pcDNA3 using restriction site HindIII in a Gibson assembly. One insert was generated by amplifying the T2A sequence using primer pair NC3354/NC3375 with pSCRPSY as a template. The second insert was made using primer pair NC3463/NC3466 with β-MAT-hp2-6m9 as a template. The resulting plasmid was used as a template for PCR amplification with primers NC2264/NC2265. This PCR product was inserted into pAAVS-EGFP-DONOR digested with FseI and XbaI by Gibson assembly to generate hAAVS1-GFP-T2A-β2-MAT-E8-3 ′hp2-6m9.

## Quantification and statistical analysis

Imagequant 5.2 was used to quantify northern blots. Bands were boxed at equal sizes in respective columns, with the rolling ball method used to subtract background. For *Figure 2B–E* and *Figure 2— figure supplement 1*, GelQuantNet was used with bands boxed at equal sizes in respective lanes and background automatically subtracted. Image Studio Ver 3.1 was used to quantify western blots. Bands were selected using 'Add Rectangle' feature with background automatically subtracted.

CRISPR screen data was analyzed by MAGeCK (see CRISPR screen methods)(*Li et al., 2014*). Poly (A)-ClickSeq was analyzed by ClickSeq Technologies using DPAC (*Elrod et al., 2019*; *Routh, 2019b*; *Routh et al., 2017*). Venn diagrams were analyzed by SuperExactTest in RStudio (*Wang et al., 2015*). All other statistical analysis performed used two-tailed, unpaired student's t-tests in Graph-Pad Prism, with mean and standard deviation displayed. When p-value not listed or ns = not significant, *p≤0.05, **p≤0.01, ***p≤0.001.

## Acknowledgements

We thank Drs. Feng Zhang, David Root, John Doench, Didier Trono, and Kathryn Pendleton for plasmids. We also thank Drs. Joshua Mendell and John Schoggins' labs for advice in designing the GFP reporter cell lines and conducting CRISPR screens. CRISPR screens were sorted with the help of the Moody Foundation Flow Cytometry Facility in the Children's Research Institute at UTSW. This research was supported by the Welch foundation I-1915–20170325 (to N.K.C), the National Institutes of Health NIGMS: R01 GM127311 to N.K.C.; R01 GM127311-S1 to J.N.F.; T32 GM007062 to A.M.S.; R35 GM136370 to B.P.T.

## Additional information

### Funding

| Funder | Grant reference number | Author |
| --- | --- | --- |
| Welch Foundation | I-1915-20170325 | Nicholas K Conrad |
| National Institute of General Medical Sciences | R01 GM127311 | Nicholas K Conrad |
| National Institute of General Medical Sciences | R01 GM127311-S1 | Juliana N Flaherty |
| National Institute of General Medical Sciences | T32 GM007062 | Anna M Scarborough |
| National Institute of General Medical Sciences | R35 GM136370 | Benjamin P Tu |

The funders had no role in study design, data collection and interpretation, or the decision to submit the work for publication.

### Author contributions

Anna M Scarborough, Conceptualization, Investigation, Methodology, Writing - original draft, Writing - review and editing; Juliana N Flaherty, Kuanqing Liu, Investigation, Methodology, Writing - review and editing; Olga V Hunter, Investigation, Methodology; Ashwani Kumar, Data curation, Methodology, Writing - review and editing; Chao Xing, Data curation, Methodology; Benjamin P Tu, Supervision, Funding acquisition, Methodology, Writing - review and editing; Nicholas K Conrad, Conceptualization, Supervision, Funding acquisition, Investigation, Methodology, Writing - original draft, Writing - review and editing

### Author ORCIDs

Anna M Scarborough (ID) https://orcid.org/0000-0003-3621-234X
Juliana N Flaherty (ID) http://orcid.org/0000-0002-9745-6762
Chao Xing (ID) http://orcid.org/0000-0002-1838-0502
Benjamin P Tu (ID) http://orcid.org/0000-0001-5545-9183
Nicholas K Conrad (ID) https://orcid.org/0000-0002-8562-0895

### Decision letter and Author response

Decision letter https://doi.org/10.7554/eLife.64930.sa1
Author response https://doi.org/10.7554/eLife.64930.sa2

## Additional files

### Supplementary files

• Supplementary file 1. Analysis of genome-wide CRISPR-Cas9 screen. MAGeCK analysis of biological replicates of CRISPR screen. Table shows MAGeCK analysis for each individual replicate (rep1-3) and analysis of the replicates together (triplicate). The triplicate analysis is referenced in text and *Figure 1*.

• Supplementary file 2. Analysis of Poly(A)-ClickSeq results. Poly(A)-ClickSeq analysis of HCT116 and 116-ΔDI cell lines with either siNon-targeting or siCFI$_m$25 treatment.

• Supplementary file 3. Nucleic acid reagents. DNA oligonucleotide sequences for cloning primers, northern blotting probe primers, RNase H oligonucleotides, CRISPR screen primers, and synthesized DNA for cloning. RNA oligonucleotides sequences for label transfer assay substrates.

• Transparent reporting form

## Data availability

Raw and unedited CRISPR screen data is deposited on GEO (GSE172217). Raw and unedited Poly (A)-ClickSeq data is deposited on GEO (GSE158591). Analysis of Poly(A)-ClickSeq is found in Supplementary File 2.

The following datasets were generated:

| Author(s) | Year | Dataset title | Dataset URL | Database and Identifier |
|---|---|---|---|---|
| Scarborough AM, Conrad NK | 2020 | NUDT21 regulates intron detention of the SAM synthetase MAT2A RNA | https://www.ncbi.nlm.nih.gov/geo/query/acc.cgi?acc=GSE158591 | NCBI Gene Expression Omnibus, GSE158591 |
| Conrad NK, Scarborough AM, Kumar A, Xing C | 2021 | CRISPR screen identifies NUDT21 as a regulator of intron detention of the SAM synthetase MAT2A RNA | https://www.ncbi.nlm.nih.gov/geo/query/acc.cgi?acc=GSE172217 | NCBI Gene Expression Omnibus, GSE172217 |

The following previously published dataset was used:

| Author(s) | Year | Dataset title | Dataset URL | Database and Identifier |
|---|---|---|---|---|
| Martin G, Gruber AR, Keller W, Zavolan M | 2012 | Genome-wide analysis of pre-mRNA 3′ end processing reveals a decisive role of human cleavage factor I in the regulation of 3′ UTR length: CLIP | https://www.ncbi.nlm.nih.gov/geo/query/acc.cgi?acc=GSE37398 | NCBI Gene Expression Omnibus, GSE37398 |

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
