## [Decision Letter]

**Acceptance summary:**

In this work, the authors extend upon their previous investigations of how the MAT2A mRNA is subject to RNA processing in response to intracellular SAM homeostasis by establishing that members of the CFI_m_ complex function in splicing of the terminal intron. The strengths of the study are the novelty that CFI_m_ subunits appear to be functioning in splicing, which has been understudied in the field. The importance of SAM homeostasis to numerous cellular processes, and the surprising implication of the polyadenylation machinery in splicing regulation contribute to the significant impact this work will have on the fields of RNA-processing and metabolism/signaling through methylation.

**Decision letter after peer review:**

Thank you for submitting your article "SAM homeostasis is regulated by CFI_m_-mediated splicing of MAT2A" for consideration by *eLife*. Your article has been reviewed by 3 peer reviewers, including Eric J Wagner as the Reviewing Editor and Reviewer #1, and the evaluation has been overseen by James Manley as the Senior Editor.

The reviewers have discussed the reviews with one another and the Reviewing Editor has drafted this decision to help you prepare a revised submission.

Summary:

In this work, the authors extend upon their previous investigations of how the MAT2A mRNA is subject to RNA processing in response to intracellular SAM homeostasis by establishing that members of the CFI_m_ complex function in splicing of the terminal intron. The strengths of the study are the novelty that CFI_m_ subunits appear to be functioning in splicing, which has been understudied in the field. Weaknesses include technical concerns revolving around how much intron detention and splicing is playing in the regulation as opposed to mRNA stability

Essential revisions:

Two main themes emerged from the discussion among the reviewers: 1) concern about the reporter as displaying a readout that is splicing dependent; 2) the desire for additional experimental support and discussion of the model presented in Figure 7. As such, the reviewers suggest the following be addressed:

1. Provide additional controls to convince the reader that the GFP reporter expression is truly dependent on its splicing efficiency. Both Reviewers 1 and 2 provide more granular suggestions on how to potentially do this below.

2. Figure 7A makes two predictions: (1) the UGUU and UGUA reporters in Figure 4G, but not the UGCU, should be responsive to CFI_m_25 knockdown and (2) co-depletion of CFI_m_25 and METTL16 should not have a bigger impact on the splicing of the endogenous MAT2A than depletion of CFI_m_25 alone. Both of these predictions are readily tested with the reagents at hand and therefore should be done.

3. Modify language in the discussion as described by Reviewer 3.

*Reviewer #1:*

1. In Figure 1, there needs to be more data provided to indicate that the GFP reporter SAM-dependence is indeed derived from splicing of detained intron in addition to mRNA-stability. The data presented are convincing that the reporter is responsive to SAM depletion (through Mat2A KD), methionine reduction. And METTL16 knock down, but the northern blots do not show the status of the DI splicing in the GFP reporter itself. The concern is that the effects seen are entirely mediated by the hp2-6. qRT-PCR would be sufficient to show this for the reporter. The authors make comments to this event later in the results stating that this product does not appear to accumulate for unknown reasons but it is concerning that it cannot be detected yet is the key component of reporter output.

2. Along these same lines, the authors describe in S1B-C that there was an unexpected splicing event in their reporter but no primary data is shown for this. This makes it really difficult to understand what is going on quantitatively at the level of splicing.

3. In Figure 2B, it appears to me that the major effect is less accumulation of the mature MAT2A mRNA (m). The authors are presenting data as 'percentage detained intron inclusion', which increases upon CFI_m_25 knock down. But the raw data seems to suggest that absolute DI levels are not changing but the mature product is going down. Could CFI_m_25 impact mRNA stability as well?

4. The CRISPR experiment to surgically remove the DI intron is clever but because the authors did not get the homozygous line they desired, I am concerned that other allele could be acting strange as it has a frameshift. Is this other allele subject to NMD? Seems the experiment would be simpler to interpret if both alleles were the same. Can the authors do a better job justifying this genetic tool?

5. If the authors can either create a homozygous δ-DI cell line or provide a reasonable response to my point #4, they should conduct the experiment shown in Figure 5C but on the δ-DI line. This experiment is slightly different than what was done in Figure 3B and is entirely more compelling because the response is robust. This would really get at the heart of what is the driving factor here: stability of the mature MAT2A mRNA or the splicing of it (that is, in response to methionine depletion).

6. The model in Figure 7 will likely confuse people. I understand that the authors are trying to keep things simple but they do not show the impact on mRNA stability of the MAT2A mRNA – which appears to be quite dominant. Moreover, the connection between METTL16 and the CFI_m_ complex has not been established in this study raising more questions. I would recommend reducing the accentuation on this component of the model.

*Reviewer #2:*

1. CFI_m_25 was initially identified at a regulator of MAT2A splicing through its strong phenotype in the initial CRISPR screen. But it is not shown that GFP expression is actually a read out of the splicing efficiency of the reporter, or that the splicing of the reporter (which is different from that observed in the endogenous MAT2A context) is dependent on SAM and/or METTL16. Without these controls it could be that CFI_m_25 knockdown impacts GFP expression through a splicing- or METTL16-independent manner.

2. Figure 3C is compelling evidence that depletion of CFI_m_25 only impacts SAM expression when the intron of MAT2A is present, but is this really due to splicing? Expression of mature MAT2A appears similarly Met responsive in the 116-DeltaD1 cells compared to HCT116 (Figure S3A – compare mature message -/+Met). Therefore an equally plausible model is that CFI_m_25 regulates the stability of the mature MAT2A?

3. The model in Figure 7A makes two predictions: (1) the UGUU and UGUA reporters in Figure 4G, but not the UGCU, should be responsive to CFI_m_25 knockdown and (2) co-depletion of CFI_m_25 and METTL16 should not have a bigger impact on the splicing of the endogenous MAT2A than depletion of CFI_m_25 alone. Both of these predictions are readily tested with the reagents at hand and therefore should be done.

*Reviewer #3:*

Rewrite the discussion. It comes off as primarily defensive of some points that are reasonable criticisms but that you have made good counter arguments for in the Results section, and does little to highlight the truly interesting and remarkable findings of the paper.

A couple of obvious but speculatory points (i.e. beyond the reasonable scope of this paper) that might be included: 1) whether canonical splicing factors that typically bind within the polypyrimidine tract (U2AF1/2) where the intronic CFI_m_25 binding site is located are bound here in MET depleted conditions, or whether they are physically and functionally replaced by the RS-domains of CFI_m_68/59. 2) Given that free METTL16 and CFI_m_25 do not show interaction it may be necessary to obtain macromolecular structures of the MAT2A DI-3UTR region/METTL16/CFI_m_ complex in order to determine the RNP interactions governing this complex formation. 3) To what degree does the role of CFI_m_ here represent a completely isolated and specific innovation by METTL16 to complete a splicing-based feedback loop, versus providing some insight into the long-understood role of 3' end formation in the splicing of the ultimate intron of a transcript?

---

## [Author Response]

Essential revisions:Two main themes emerged from the discussion among the reviewers: 1) concern about the reporter as displaying a readout that is splicing dependent; 2) the desire for additional experimental support and discussion of the model presented in Figure 7. As such, the reviewers suggest the following be addressed:1. Provide additional controls to convince the reader that the GFP reporter expression is truly dependent on its splicing efficiency. Both Reviewers 1 and 2 provide more granular suggestions on how to potentially do this below.

The reviewer’s fairly point out that several shortcomings of our GFP reporter obscured whether the regulation by CFI_m_25 is due to splicing activity or to changes in RNA stability. These concerns were based primarily on three results: 1) We did not observe the detained intron isoform (GFP-DI) in our northern blots, so it was questionable whether it was even regulated by intron detention, 2) the reporter spliced in an unexpected pattern (Figure 1—figure supplement 1), and 3) we did not exclude potential roles of CFI_m_25 in hp2-6 regulation of mRNA stability. We have developed two new reagents that we think rigorously address these concerns and strongly support the conclusion that the GFP reporters (and MAT2A) are regulated by CFI_m_25 at the level of splicing of the detained intron.

First, we have improved the quality of our northern blot GFP probe. In the previous version of the manuscript, we mentioned that we could not visualize the isoform of GFP containing the detained intron (GFP-DI). In fact, we did not even observe the mRNA except under inducing (-methionine) conditions (see Figure 1D and Figure 1E). Since then, we designed a significantly improved GFP probe that detects both isoforms of our reporter (revised Figure 2B and Figure 2—figure supplement 1). To improve the signal-to-noise in the assay even more, we also added a poly(A)-selection step to our northern blot validation experiments in Figure 2B-2E and Figure 2—figure supplement 1. To address Reviewer #1’s concern regarding quantification, we included quantification of these data not only as %DI, but also as relative levels of each of the isoforms (Figure 2C-2E). Using the original reporter, we see concomitant decreases of mRNA with increases in the GFP-DI isoform upon knockdown of METTL16 or CFI_m_25. These observations support the proposed roles of CFI_m_25 and METTL16 in the regulation of splicing of the detained intron of the reporter used in the CRISPR screen.

We have kept the data with the old probe in Figures 1D and 1E. We could repeat those experiments with the new GFP probe to produce a cleaner result, but that seems misleading since our screen was rationalized based on those data.

Second, we have included data from a modified reporter recently developed in the lab (revised Figure 2A, bottom diagram). The reporter is effectively the same as the one used previously except for two changes. This reporter includes a T2A “self-cleaving” peptide between the GFP and the β-globin MAT2A fusions for better protein stability. (The stability issue was described in the original version and remains in the revised Results section and Figure 1—figure supplement 1.) However, the relevant change for the current paper is that the new reporter has hp2-6 mutated, so it is not subject to regulation by hp2-6-mediated RNA stability. Using this reporter (T2A, hp2-6m9), we observed diminished GFP mRNA levels and increased GFP-DI accumulation upon depletion of METTL16 or CFI_m_25 (Figure 2A-E; Figure 2—figure supplement 1). A similar response was observed with two independently derived integrated clonal cell lines (Figure 2B-E and Figure 2—figure supplement 1). This observation excludes the possibility that the CFI_m_25 regulation is hp2-6 dependent and is consistent with a role for these factors in splicing of the MAT2A detained intron in the GFP reporter.

As a side note, careful inspection of the northern blots in revised Figure 2B show that the new T2A, hp2-6m9 reporter mRNA is slightly longer and the detained intron isoform is slightly shorter than the original “Reporter” counterparts. These patterns are consistent with the fact that the new reporter lines’ splicing patterns reflect the “predicted” pattern discussed in Figure 1—figure supplement 1, while the original reporter is distinct as previously reported. Thus, the effects of CFI_m_25 on the reporter is not specific to the unique splicing pattern in the original reporter.

With these new data, we have specifically demonstrated that CFI_m_25-mediated regulation of our MAT2A reporter does not require hp2-6. Conversely, both the initial submission and the revised version of the paper include data from the 116-ΔDI line that demonstrates that CFI_m_25 regulation of the endogenous MAT2A requires the detained intron (Figure 3A-3C; Figure 3—figure supplement 1A). Since hp2-6 are not required for CFI_m_25 regulation but the detained intron is, these data strongly support the conclusion that the effect of CFI_m_25 on GFP mRNA accumulation is due to a novel role for CFI_m_25 in splicing of the MAT2A detained intron.

2. Figure 7A makes two predictions: (1) the UGUU and UGUA reporters in Figure 4G, but not the UGCU, should be responsive to CFI_m_25 knockdown and (2) co-depletion of CFI_m_25 and METTL16 should not have a bigger impact on the splicing of the endogenous MAT2A than depletion of CFI_m_25 alone. Both of these predictions are readily tested with the reagents at hand and therefore should be done.

These are excellent suggestions, and we have included both of these experiments in the revised manuscript. To address the second point first, we have now included METTL16/CFI_m_25 co-depletion experiments in revised Figure 2H. As predicted by the model, there is no synergistic/additive effect after co-depletion of both CFI_m_25 and METTL16 on endogenous MAT2A RNA isoforms.

We also include the requested experiment with the mutant reporters in revised Figure 4—figure supplement 1. As predicted by the reviewer, the UGUA reporter responds similarly to wild-type upon either CFI_m_25 or METTL16 knockdown, but the UGCU is unaffected. However, there is an important technical caveat regarding the behavior of the UGCU mutant. In our hands, siRNA knockdowns are considerably more efficient in the 293A derivative line 293A-TOA than in HEK293 cells. The latter were used for the reporter experiments in Figure 4, but since the requested experiments required knockdown, we used 293A-TOA cells for the requested experiments. However, 293A-TOA cells show a much higher baseline intron detention than HEK293 cells. As with many alternative splicing events, we find considerable variability in intron retention patterns with both endogenous and reporter RNAs in different cell lines (e.g. Park et al. Cell Reports 2017). In the case that the counter-hypothesis were true (i.e. siCFI_m_25 has additive effects with UGCU), we would not be able to see this change because there is little dynamic range to increase intron retention. Therefore, these data remain largely inconclusive on this issue. These caveats are included in the legend to Figure 4—figure supplement 1.

3. Modify language in the discussion as described by Reviewer 3.

We have updated the Discussion to include the points raised by Reviewer #3.